

# Importance of microphysical settings for climate forcing by stratospheric $SO_2$ injections as modelled by SOCOL-AERv2

Sandro Vattioni[1,*], Andrea Stenke[1,2,3,*], Beiping Luo[1], Gabriel Chiodo[1], Timofei Sukhodolov[4], Elia Wunderlin[1], and Thomas Peter[1]

[1]Institute for Atmospheric and Climate Science, ETH Zurich, Switzerland
[2]Institute of Biogeochemistry and Pollutant Dynamics, ETH Zurich, Switzerland
[3]Eawag, Swiss Federal Institute of Aquatic Science and Technology, Dübendorf, Switzerland
[4]Physikalisch-Meteorologisches Observatorium Davos and World Radiation Center, Davos, Switzerland
[*]These authors contributed equally to this work.

**Correspondence:** Sandro Vattioni (sandro.vattioni@env.ethz.ch), Andrea Stenke (andrea.stenke@env.ethz.ch)

**Abstract.** Solar radiation management as a sustained deliberate source of $SO_2$ into the stratosphere (strat-SRM) has been proposed as an option for climate intervention. Global interactive aerosol-chemistry-climate models are often used to investigate the potential cooling efficiencies and side effects of hypothesised strat-SRM scenarios. A recent strat-SRM model intercomparison study for composition-climate models with interactive stratospheric aerosol suggests that the modelled climate response to a particular assumed injection strategy, depends on the type of aerosol microphysical scheme used (e.g., modal or sectional representation), alongside also host model resolution and transport. Compared to short-duration volcanic $SO_2$ emission, the continuous $SO_2$ injections in strat-SRM scenarios may pose a greater challenge to the numerical implementation of of microphysical processes such as nucleation, condensation, and coagulation. This study explores how changing the timesteps and sequencing of microphysical processes in the sectional aerosol-chemistry-climate model SOCOL-AERv2 (40 size bins) affect model predicted climate and ozone layer impacts considering strat-SRM $SO_2$ injections of of 5 and 25 Tg(S) yr$^{-1}$ at 20 km altitude between 30°S and 30°N. The model experiments consider year 2040 boundary conditions for ozone depleting substances and green house gases. We focus on the length of the microphysical timestep and the call sequence of nucleation and condensation, the two competing sink processes for gaseous $H_2SO_4$. Under stratospheric background conditions, we find no effect of the microphysical setup on the simulated aerosol properties. However, at the high sulfur loadings reached in the scenarios injecting 25 Mt/yr of sulfur with a default microphysical timesetp of 6 min, changing the call sequence from the default "condensation first" to "nucleation first" leads to a massive increase in the number densities of particles in the nucleation mode ($R < 0.01$ μm) and a small decrease in coarse mode particles ($R > 1$ μm). As expected, the influence of the call sequence becomes negligible when the microphysical timestep is reduced to a few seconds, with the model solutions converging to a size distribution with a pronounced nucleation mode. While the main features and spatial patterns of climate forcing by $SO_2$ injections are not strongly affected by the microphysical configuration, the absolute numbers vary considerably. For the extreme injection with 25 Tg(S) yr$^{-1}$, the simulated net global radiative forcing ranges from -2.3 W m$^{-2}$ to -5.3 W m$^{-2}$, depending on the microphysical configuration. "Nucleation first" shifts the size distribution towards radii better suited for solar scattering (0.3 μm $< R <$ 0.4 μm), enhancing the intervention efficiency. The size-distribution shift however generates more



ultra-fine aerosol particles, increasing the surface area density, resulting in 10 DU less ozone (about 3% of total column) in the northern midlatitudes and 20 DU less ozone (6%) over the polar caps, compared to the "condensation first" approach. Our results suggest that a reasonably short microphysical time step of 2 minutes or less must be applied to accurately capture the magnitude of the $H_2SO_2$ supersaturation resulting from $SO_2$ injection scenarios or volcanic eruptions. Taken together these results underscore how structural aspects of model representation of aerosol microphysical processes become important under conditions of elevated stratospheric sulfur in determining atmospheric chemistry and climate impacts.

# 1  Introduction

The idea of increasing the Earth's albedo by injecting sulfur containing gases into the stratosphere to reduce some of the adverse effects of greenhouse-gas (GHG) induced global warming dates back to the 1970s (Budyko, 1974), and was 30 years later further elaborated by Crutzen (2006). The arguments presented by Crutzen called for active scientific research of the kind of activity, which became known under the somewhat misleading term "geoengineering". We term this here "climate intervention", following the recommendation of the National Research Council (2015). Crutzen's idea is based on the fact that sulfur containing gases, such as $SO_2$, $H_2S$ or OCS, injected into the lower stratosphere will form aqueous sulfuric acid aerosol particles via a chain of chemical and microphysical processes (Thomason and Peter, 2006; Kremser et al., 2016). The resulting binary $H_2SO_4$-$H_2O$ solution droplets reflect solar radiation back to space, causing a cooling at the Earth's surface. At the same time, however, they heat the stratosphere due to absorption of upwelling long-wave radiation. Moreover, sulfate aerosols play an important role in stratospheric ozone chemistry by providing surfaces for heterogeneous reactions (Solomon, 1999). While the infrared absorptivity is determined to good approximation by the total aerosol volume, the efficiency of scattering solar radiation depends strongly on the detailed aerosol size distribution: Many small particles are more efficient than a few large particles, but they also provide a larger surface area density (SAD) accelerating heterogeneous chemistry (Heckendorn et al., 2009).

In the stratosphere, the total aerosol number density and size distribution are governed by the microphysical processes of nucleation, coagulation, condensation, evaporation, and gravitational settling (Kremser et al., 2016, and references therein). The formation of new sulfate aerosol particles occurs via binary homogeneous nucleation of $H_2SO_4$ and $H_2O$ molecules, or, via heterogeneous nucleation in the presence of appropriate condensation nuclei like meteoritic dust or ions, which requires lower saturation ratios than homogeneous nucleation. The freshly formed particles can grow further through coagulation as well as condensation of $H_2SO_4$ (together with $H_2O$). As stratospheric temperatures increase with altitude, the sulfate aerosol particles eventually evaporate above 32 to 35 km, releasing $H_2SO_4$ back to the gas phase.

The effectiveness of climate intervention by $SO_2$ emission has been intensively investigated by using models of different complexity and assuming different injection scenarios (e.g., Heckendorn et al., 2009; Pierce et al., 2010; Niemeier et al., 2011; English et al., 2011; Niemeier and Timmreck, 2015; Tilmes et al., 2018; Vattioni et al., 2019; Weisenstein et al., 2022; Laakso et al., 2022; Visioni et al., 2022). Such modelling studies have advanced our understanding of stratospheric aerosols, but they also highlighted uncertainties regarding the transport, chemistry, and microphysics of the aerosol size distribution.



In a recent study, Weisenstein et al. (2022) presented a model intercomparison exploring the impacts of stratospheric injections of $SO_2$ gas as well as accumulation-mode sulfuric acid aerosol (AM-$H_2SO_4$) on atmospheric chemistry and climate. Three general circulation models (GCMs) with interactive aerosol microphysics conducted strictly coordinated model experiments within the framework of the Geoengineering Model Intercomparison Project (GeoMIP, Kravitz et al., 2011), namely the Community Earth System Model (CESM2) with the Whole Atmosphere Community Climate Model (WACCM) atmospheric configuration (Danabasoglu et al., 2020), the middle atmosphere version of ECHAM5 with the HAM microphysical module (MAECHAM5-HAM; Stier et al., 2005), and the SOlar Climate Ozone Links model with AER microphysics (SOCOL-AER) version 2 (Feinberg et al., 2019). The model experiments included injections of 5 and 25 $Tg(S)\,yr^{-1}$ in form of $SO_2$ gas or AM-$H_2SO_4$, emitted either as two point injections at 30°N and 30°S or as regional injection between 30°N and 30°S. Two of the participating models, CESM2 and MAECHAM5-HAM, assume the aerosol size distribution can be described by superimposed lognormal size distributions (modal scheme), while SOCOL-AERv2 uses a size bin-resolving (sectional) scheme.

The analysis of the simulated particle size distributions for the $SO_2$ injection scenarios revealed substantial differences between each pair of the three models. CESM2 generates new particles and adds them directly to the Aitken mode ($R \gtrsim 10\,nm$), so that there are no nm-sized particles. In contrast, SOCOL-AERv2 treats these tiny particles down to 0.4 nm. Compared to MA-EACHM-HAM, SOCOL shows substantially fewer nucleation mode particles, suggesting different roles of nucleation and condensation in both models: the microphysical scheme in SOCOL-AERv2 appears to prefer condensational growth of existing particles by uptake of $H_2SO_4$ over the formation of new particles, while the opposite seems to be the case for MAECHAM5-HAM. The description of the results of the microphysical processes by means of lognormal functions in modal models, such as CESM2 and MAECHAM5-HAM, further complicates the interpretation.

Nucleation and condensation are competing sink processes for gas-phase $H_2SO_4$, which occur simultaneously in the atmosphere, but typically with different speeds. The characteristic time scale $\tau$ for removal of $H_2SO_4$ molecules by condensation is given by the following equation:

$$\tau_{cond} = \frac{4}{Av}, \tag{1}$$

with $A$ being the aerosol surface area density and $v$ the mean thermal velocity of $H_2SO_4$ molecules. For background conditions with typical SAD values of 5 to 10 $\mu m^2 cm^{-3}$ in nucleation regions, the equilibrium time scale for condensation is around 0.5–1 h. This value decreases inversely with increasing SAD. Under volcanic or geoengineered conditions with SAD $\sim$ 80 $\mu m^2 cm^{-3}$, typical for the 25 $Tg(S)\,yr^{-1}$ injection scenario, the equilibrium time scale is less than 5 minutes. As the nucleation rate strongly depends on the gas-phase $H_2SO_4$ supersaturation, the model timestep used for condensation and nucleation must be significantly smaller than the time required to approach gas-phase equilibrium in order to avoid that one process erroneously dominates the gas-to-particle transfer of $H_2SO_4$. Furthermore, coagulation is also affected by the competition between nucleation and condensation, as it is most efficient at (initially) high number densities and between particles of different size. Small particles move fast, but have only small cross-sections for collision, while large particles have a slower Brownian motion, but provide good collision targets for smaller particles (Seinfeld and Pandis, 1997). The correct numerical representation of these simultaneously occurring processes is challenging, especially under sulfur-rich conditions, when characteristic time



scales become extremely short. This motivated us to critically question the microphysical scheme of the sectional SOCOL-AERv2 model and to systematically test the impact of the call sequence of the subroutines for condensation and nucleation, as well as the microphysical timestep on the simulated aerosol properties and the modeled climate response to stratospheric $SO_2$ injection.

The paper is organised as follows: Section 2 presents a brief description of the SOCOL-AERv2 model and details of the experimental setup. Section 3 discusses the impact the microphysical settings on the aerosol size distribution under stratospheric background conditions as well as under stratospheric injections of $SO_2$ gas (3.1), on the global mean particle size, aerosol burden and radiative forcing (3.2), and on the meridional distributions of aerosol burden, radiative forcing, and ozone (3.3) resulting from the $SO_2$ injections. The influence of the microphysical settings on profiles of various quantities is briefly

mentioned (3.4) and detailed in the Supplement. To evaluate the changes in SOCOL aerosol microphysics against observations we also tested different settings for the 1991 eruption of Mt. Pinatubo (3.5). Section 4 includes a summary and discussion.

## 2 Model description and experimental setup

### 2.1 SOCOL-AERv2

A first version of the aerosol-chemistry-climate model SOCOL-AER had been introduced by Sheng et al. (2015), who inte-

grated the size-resolving (sectional) sulfate aerosol module AER (Weisenstein et al., 1997) into the three dimensional grid of the chemistry-climate model (CCM) SOCOLv3 (Stenke et al., 2013), which consists of the middle atmosphere version of the spectral general circulation model MA-ECHAM5 (Roeckner et al., 2003, 2006) and the chemistry-transport model MEZON (Rozanov et al., 1999; Egorova et al., 2003). Since then, the model's tropospheric and stratospheric sulfur cycle have undergone several improvements, resulting in the publication of SOCOL-AERv2 (Feinberg et al., 2019).

SOCOL-AERv2 resolves the sulfate aerosol particles in 40 size bins, ranging from 0.39 nm to 3.2 μm in radius. Since the size bins refer to dry aerosol radius, they can also be interpreted as aerosol $H_2SO_4$ mass bins, ranging from about 2.8 molecules to $1.6 \times 10^{12}$ molecules of $H_2SO_4$ per aerosol particle. Neighboring size bins differ by molecule number doubling.

  Detailed descriptions of the original AER microphysics and their adaptations for the coupled model are provided in Weisenstein et al. (1997, 2007) and Sheng et al. (2015), respectively. Aerosol composition, i.e. the sulfuric acid weight percent in the

particles, is calculated as function of ambient temperature and $H_2O$ partial pressure using the parameterization of Tabazadeh et al. (1997), which is also used for the calculation of the wet aerosol radius of each size bin. For the formation of new particles by binary homogeneous nucleation the scheme of Vehkamäki et al. (2002) is used. The scheme calculates the nucleation rate as well as the radius and composition of new particles, meaning that the nucleated mass is added to a single size bin. The particles can grow through $H_2SO_4$ condensation and shrink through evaporation, both processes depending on the equilibrium

concentration of $H_2SO_4$ above the particle surface (Ayers et al., 1980; Kulmala and Laaksonen, 1990). Condensational growth leads to an increase of mass in the aerosol phase and a shift of particles to larger size bins, while evaporation does the opposite. Changes in the net number density occur only upon evaporation from the smallest size bin or condensational growth of the largest size bin. Finally, coagulation reduces number densities and shifts aerosol mass to larger bins. Coagulation is solved by a





semi-implicit method (Jacobson and Seinfeld, 2004), whereas at most 90% of the available mass in one size bin is allowed to be
lost by coagulation within one microphysical time step. Otherwise, the coagulation timestep is reduced. The coagulation ker-
nel, which defines the collision probability of two particles, depends on the particle radius and the diffusion coefficient (Fuchs,
1964). Finally, sedimentation, which affects the vertical distribution of aerosol particles and reduces their residence time in
the stratosphere, is parameterised following the numerical scheme of Walcek (2000). The gravitational settling velocities of
aerosol particles are calculated following Kasten (1968).

The CCM SOCOLv3 and the aerosol module AER are interactively coupled via the chemistry and radiation routines. Sulfur
chemical reactions (Sheng et al., 2015, see Table 1) are fully integrated into the model's chemical solver, which is based on the
implicit iterative Newton-Raphson scheme (Stott and Harwood, 1993). In addition to gas-phase chemistry, the model includes
aqueous-phase oxidation of S(IV) to S(VI) by ozone ($O_3$) and hydrogen peroxide ($H_2O_2$) in cloud water (Jacob, 1986).
The modeled sulfate aerosol is fed directly into the heterogeneous chemistry and radiation schemes. The aerosol radiative
properties (extinction coefficients, single scattering albedos, and asymmetry factors as functions of wavelength) required to
drive the model dynamics are calculated online from the aerosol size distribution using Mie theory with a temperature- and
humidity-dependent look-up table, which accounts for the aerosol $H_2O_2$ weight percent.

The model uses operator splitting. The dynamics module is called every 15 min, whereas the chemistry, aerosol micro-
physics, and radiation schemes are called every 2 h. For the microphysical processes, especially for nucleation with its highly
nonlinear dependence on gaseous $H_2SO_4$ concentration, sub-timesteps are used within the 2-h chemistry loop, to avoid that the
process called first mistakenly dominates the $H_2SO_4$-to-particle exchange rate. The default procedure is to use $N_{micro} = 20$
sub-loops within the chemical timestep, which results in a microphysical timestep of 6 minutes (2 h/$N_{micro}$ = 2 h/20 = 6 min).
The parameter $N_{micro}$ can be easily adjusted between runs. By default, the call sequence for the microphysical processes is
condensation first, followed by nucleation (see "CN" sequence in Fig. 1), and finally coagulation. At each chemical timestep,
the model first calculates the new $H_2SO_4$ gas-phase concentration resulting from chemical production and transport. In the mi-
crophysical loop, the $H_2SO_4$ concentration is then consecutively updated by condensation and nucleation. As we will see later,
it is also important to distribute the gaseous $H_2SO_4$ molecules produced during the 2-h chemical timestep homogeneously over
the $N_{micro}$ sub-timesteps (see $\Delta H_2SO_4$ in Fig. 1), rather than passing them as a total amount at the beginning of the micro-
physical loop. Otherwise, under conditions of $SO_2$ injections the 2-hourly call frequency of the chemistry scheme would lead
to initially unrealistically high $H_2SO_4$ supersaturations in the microphysical loop, which then causes artefacts in the aerosol
microphysics. The aerosol composition as well as the coagulation kernel are calculated only once every 2 h and are kept con-
stant for the microphysical calculations. Finally, sedimentation is calculated after the microphysical subloop, again once every
2 h. To test the implications of the aerosol microphysics on the simulated aerosol size distribution under various stratospheric
sulfur loadings, we performed several sensitivity simulations, for which we changed the call sequence for condensation and
nucleation or increased the number of microphysical sub-timesteps $N_{micro} > 20$.



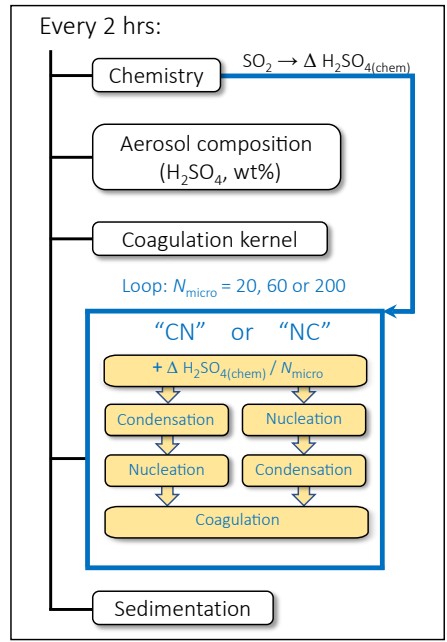

**Figure 1.** Schematic representation of the call sequence for the microphysical processes in SOCOL-AERv2. The scheme shows the setup with the microphysical subloop with $N_{\mathrm{micro}} = 20$ steps by default, or an increased number of steps (60, 80, or 200) in the test runs. By default, the condensation routine is called first and nucleation second ("CN"), which was reversed ("NC") for the tests. Furthermore, the amount of $H_2SO_2$ gas molecules produced by the chemistry scheme is uniformly distributed over the $N_{\mathrm{micro}}$ time steps, instead of providing the total amount at the first microphysical time step, as done in the original set-up.

## 2.2 SOCOLv4

Additionally, we performed simulations with the fully coupled Earth System Model (ESM) SOCOLv4 (Sukhodolov et al., 2021), which is a further development of SOCOL-AERv2. SOCOLv4, incorporates the same aerosol module, AER, as SOCOL-AERv2 (Sect. 2.1). The major differences between the model versions is that SOCOLv4 is based on the MPI-ESM1.2 (Mau-
ritsen et al., 2019), which incorporates the fully coupled interactive ocean module MPIOM (Jungclaus et al., 2013). SOCOLv4 has a finer horizontal and vertical resolution with T63 truncation (1.9° x 1.9°) and 47 vertical pressure levels also reaching up to 0.01 hPa. Compared to SOCOL-AERv2 the default dynamical timestep was halved to 7.5 min, while the default chemical and microphysical time steps are the same as for SOCOL-AERv2 (2h and 6 min, respectively). The interactive ocean as well as the finer spatial resolution make SOCOLv4 computationally much more expensive than SOCOL-AERv2. Therefore, we
performed most sensitivity simulations with SOCOL-AERv2 using fixed sea surface temperatures (SST) and sea ice coverage (SIC, see section2.3), while SOCOLv4 was primarily used to look at the impact on surface temperature anomalies.



## 2.3 Experimental setup

For the present study, we employed SOCOL-AERv2 with a resolution of 39 hybrid sigma-pressure levels in the vertical and a horizontal trunction of T42 ($\sim 2.8° \times 2.8°$ in latitude and longitude). The simulations for this study include a reference
scenario for stratospheric background conditions as well as two perturbation scenarios including stratospheric sulfur injections. The boundary conditions are identical to the GeoMIP test-bed experiment "accumH2SO4"[1] with GHGs and ozone-depleting substances taken from the projections for 2040 from the SSP5-8.5 scenario (see also Weisenstein et al. (2022)). SST and SIC are prescribed using an average of the years 1988–2007 of the CMIP5 PCMDI-AMIP-1.1.0 SST/Sea Ice dataset (Taylor et al., 2000). As SOCOL-AERv2 with 39 vertical levels does not generate a quasi-biennial oscillation (QBO) internally, the
simulated wind in the equatorial stratosphere is nudged towards observed wind profiles (Stenke et al., 2013). We ran 20 model years for each scenario. The first 5 years are considered as spin-up period (sufficient for the present application), and we use the subsequent 15 years for our analysis.

Consistent with Weisenstein et al. (2022), the intervention scenarios examined here apply gaseous $SO_2$ injections of 5 and 25 $\mathrm{Tg(S)\,yr^{-1}}$ emitted uniformly in a 2 km thick layer centred around 20 km altitude in the region between 30°S and 30°N
over all longitudes. These so-called "regional injections" are complemented by an example of a "point injection" performed with SOCOLv4 (see section 2.2) injecting 5 $\mathrm{Tg(S)\,yr^{-1}}$ of $SO_2$ at the same vertical extent but constrained to a region from 10°N to 10°S at the equator only emitting at the 0° meridian. These point emission scenarios followed the G4 GeoMIP scenario (Kravitz et al., 2011) with transient SSP5-8.5 boundary conditions and allow us to explore the sensitivity of surface temperature to the call sequence in a fully coupled ESM.

To determine the effects of the setup of the microphysical scheme (see Fig. 1) on the computed size distribution and aerosol burden, we performed several model simulations for background conditions as well as conditions of climate intervention. The different simulations vary by reversing the call sequence of the condensation and nucleation routines, or by increasing the number of microphysical timesteps $N_{\mathrm{micro}}$. The model simulations are summarized in Table 1. The experiment BG_CN_20 represents the default setup of the microphysical scheme in SOCOL-AERv2 and is used as the reference simulation.

In the absence of observational data of the stratospheric aerosol layer under geoengineering conditions, we also tested the effect of different microphysical settings in the modeling of the 1991 Mt. Pinatubo eruption, following the experimental setup of the Interactive Stratospheric Aerosol Model Intercomparison Project (ISA-MIP, Quaglia et al., 2023).

## 3 Results

In this section, we first analyze how the microphysical settings in SOCOL-AERv2 affect the calculated aerosol size distributions
under stratospheric background conditions and under scenarios with $SO_2$ injection. Next, we examine how the changes in size distributions affect global aerosol properties, such as aerosol loading and net radiative forcing. Finally, we show that

---

[1]Details of the experiment protocol: http://climate.envsci.rutgers.edu/geomip/testbed.html





**Table 1.** Overview of model simulations performed with SOCOL-AERv2 (except for S5p, which was performed with SOCOLv4). Simulation names refer to the following naming convention: "SO$_2$ emission scenario"_"Call sequence"_"$N_{micro}$". BG: background; S5: 5 Tg(S)yr$^{-1}$, regional emission; S5p: 5 Tg(S)yr$^{-1}$, point emission simulated with SOCOLv4; S25: 25 Tg(S)yr$^{-1}$, regional emission; PIN: Pinatubo eruption ("shallow injection" scenario of ISA-MIP (Timmreck et al., 2018)); CN: condensation first; NC: nucleation first; $N_{micro}$: number of microphysical timesteps.

| Simulation name | SO$_2$ emission scenario | Microphysical call sequence | Microphysical timesteps |
|---|---|---|---|
| BG_CN_20 | background | Cond-Nuc | 20 |
| BG_NC_20 | | Nuc-Cond | 20 |
| S5_CN_20 | 5 Tg(S) yr$^{-1}$ | Cond-Nuc | 20 |
| S5_CN_200 | (regional) | Cond-Nuc | 200 |
| S5_NC_20 | | Nuc-Cond | 20 |
| S5_NC_200 | | Nuc-Cond | 200 |
| S5p_CN_20 | 5 Tg(S) yr$^{-1}$ | Cond-Nuc | 20 |
| S5p_NC_20 | (point) | Nuc-Cond | 20 |
| S25_CN_20 | 25 Tg(S) yr$^{-1}$ | Cond-Nuc | 20 |
| S25_CN_200 | (regional) | Cond-Nuc | 200 |
| S25_NC_20 | | Nuc-Cond | 20 |
| S25_NC_60 | | Nuc-Cond | 60 |
| S25_NC_200 | | Nuc-Cond | 200 |
| PIN_CN_20 | Pinatubo 5 Tg(S) | Cond-Nuc | 20 |
| PIN_NC_20 | (single event, point) | Nuc-Cond | 20 |
| PIN_NC_200 | | Nuc-Cond | 200 |

microphysical settings directly affect stratospheric chemistry and thus the ozone layer via aerosol surface area density under conditions with climate intervention.

### 3.1 Influence of microphysical settings on aerosol size distribution

The upper row of panels in Fig. 2 shows particle size distributions at 55 hPa in the low latitudes (30°S-30°N) for unperturbed and for conditions with climate intervention. As obvious from panel (a), changing the call sequence of the nucleation and condensation subroutines does not influence the simulated aerosol size distribution under background conditions. Since maximum nucleation rates occur about 2-3 km below the tropical tropopause (Thomason and Peter, 2006), we also examined the size distributions at 115 hPa (not shown), and again find that the call sequence has no impact on the model results. This indicates that



the default microphysical timestep of 6 min is sufficiently shorter than the characteristic times of nucleation and condensation under background conditions, so that none of the two processes inappropriately dominates the $H_2SO_4$-to-particle conversion.

In contrast to background conditions, the $SO_2$ injections scenarios are highly sensitive to the microphysical settings. Initially, we kept the microphysical timestep constant ($N_{micro} = 20$), but reversed the call sequeNCe from the default "condensation first" (CN) to "nucleation first" (CN). This modification leads to a massive increase of nucleation mode particles ($R < 0.01$ μm)

(Fig. 2c, e, yellow and blue dotted lines).

To highlight differences in the coarse mode ($R > 1$ μm), we calculated the fifth moment of the corresponding size distributions (lower row in Fig. 2). This provides an estimate of the downward mass flux due to aerosol sedimentation, which is determined by the product of particle volume (proportional to the third moment) and sedimentation speed (roughly proportional to the second moment). Swapping from CN to NC leads to a significant decrease of coarse mode particles (by one order

of magnitude) for the S25 scenario (inset in Fig. 2f).

These significant differences in the size distributions demonstrate the dominating role of the first-called process as $H_2SO_4$ sink, either condensation or nucleation, indicating that the default timestep (2 h/$N_{micro}$ = 2 h/20 = 6 min) is too long to properly handle elevated stratospheric sulfur loadings. Therefore, we increased the number of microphysical substeps until the resulting particle size distributions of the CN and CN simulations converge. For a sufficiently short microphysical timestep (0.6 min

with $N_{micro} = 200$), the simulations develop a pronounced peak of nucleation mode particles (Fig. 2c, e, orange and blue solid lines) similar to the CN_20 simulations, but with somewhat lower particle number densities.

As expected, the computational costs of the model increase with a shorter microphysical timestep. Increasing the number of microphysical substeps from 20 to 200 almost doubles the required wall-clock time per model year from 4.6 h to 9 h, using parallel computing on 64 CPUs. To assess possibilities to reduce the computational costs, we tested the efficiency of

$N_{micro} = 60$ (and 80, not shown).

The red lines in Fig. 2e,f show the results for S25_CN_60, demonstrating excellent agreement with $N_{micro} = 200$, which gives us confidence in the accuracy of the model solution. Furthermore, the computational demand increased only moderately by about 33% (60 min) per model year (relative to $N_{micro} = 20$). In conclusion, in SOCOL-AERv2 nucleation first with $N_{micro} = 60$ provides a very good description of climate intervention scenarios, even when the loading is extremely high.

We also explored the effects of the distribution of gaseous $H_2SO_4$ molecules produced during the 2-hourly call of the chemistry routine, either homogeneously across the $N_{micro}$ sub-timesteps or as a total amount at the beginning of the microphysical loop. As Fig. S1 in the Supplement shows, proper partitioning of the $H_2SO_4$ molecules among the $N_{micro}$ sub-timesteps is critical to avoid an excessive formation of nucleation mode particles due to artificially high $H_2SO_4$ supersaturations at the beginning of the microphysical substepping. More details can be found in the supplementary material (see Section S2).

## 3.2 Influence of microphysical settings on global means of particle size, aerosol burden and radiative forcing

The large differences in the simulated size distribution have wide implications for other key metrics of stratospheric aerosols, namely the average size of the aerosol particles, burden and radiative forcing: these are collectively shown in Fig. 3 in the three sets of experiments. Figure 3a shows the globally averaged effective radius ($R_{eff}$) at 55 hPa. For background conditions





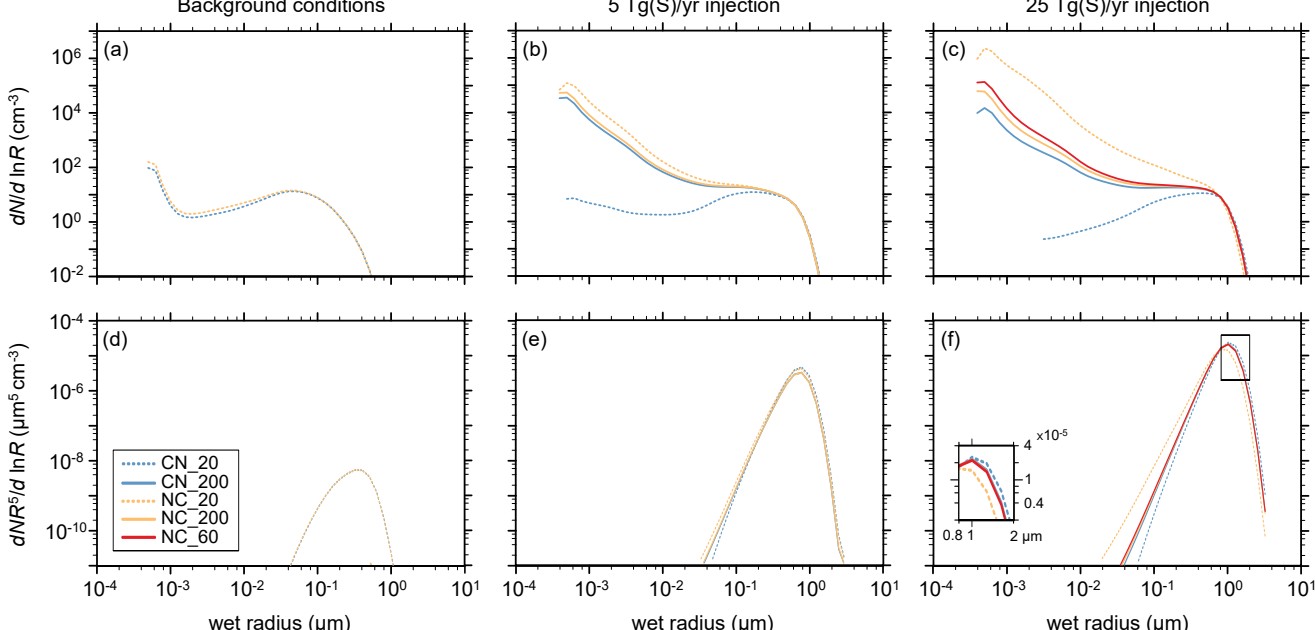

**Figure 2.** Upper row: Size distributions ($dN/d\ln R$, particles $\mathrm{cm}^{-3}$) averaged between 30°S and 30°N at 55 hPa for the model simulations with (**a**) stratospheric background conditions, (**c**) regional $SO_2$ injections of 5 $\mathrm{Tg(S)\,yr}^{-1}$, and (**e**) 25 $\mathrm{Tg(S)\,yr}^{-1}$. Lower row: The fifth moment ($dNR^5/d\ln R$, $\mathrm{\mu m}^5\,\mathrm{cm}^{-3}$) of the aerosol size distributions as an estimate for aerosol sedimentation mass flux (particle volume ($\propto R^3$) times sedimentation velocity ($\propto R^2$)). Blue lines: simulations with condensation first; orange and red lines: nucleation first. Dashed lines: $N_\mathrm{micro} = 20$ microphysical timesteps; solid orange and blue lines: $N_\mathrm{micro} = 200$; solid red lines in (e) and (f): $N_\mathrm{micro} = 60$. Insert in (f) highlights the differences for coarse particles.

both microphysical settings, CN and CN, result in an average $R_\mathrm{eff}$ of 0.15 µm. For the $SO_2$ injections scenarios, most of the
additional sulfur condenses onto existing particles or is consumed in nucleation of new particles, which coagulate preferentially onto the larger background particles. This increases the simulated $R_\mathrm{eff}$ compared to the background case, moving towards and into the range of optimal effective radius for scattering of solar radiation between 0.3 and 0.4 µm (Weisenstein et al., 2022, see their Fig. 4). The standard microphysical setup (CN, $N_\mathrm{micro} = 20$, solid blue circles) result in the largest simulated $R_\mathrm{eff}$, as condensation partly suppresses the subsequent formation of smaller particles via nucleation. Conversely, nucleation first with
long microphysical timesteps (CN, $N_\mathrm{micro} = 20$, solid orange circles) exaggerates the formation of small particles, resulting in an underestimation of $R_\mathrm{eff}$. Given a sufficiently short timestep ($N_\mathrm{micro} = 200$), CN and NC converge to $R_\mathrm{eff}$ of 0.38 µm for the S5 scenario, and 0.48 µm for the S25 scenario (blue and orange crosses). Compared to the modal models MAECHAM5-HAM and CESM2, the sectional SOCOL-AER in general produces smaller $R_\mathrm{eff}$ for the regional injections. Hence, improving the SOCOL-AER aerosol microphysics by swapping the sequence to nucleation first and increasing $N_\mathrm{micro}$ leads to a slight
reduction in the spread in $R_\mathrm{eff}$ among these models.





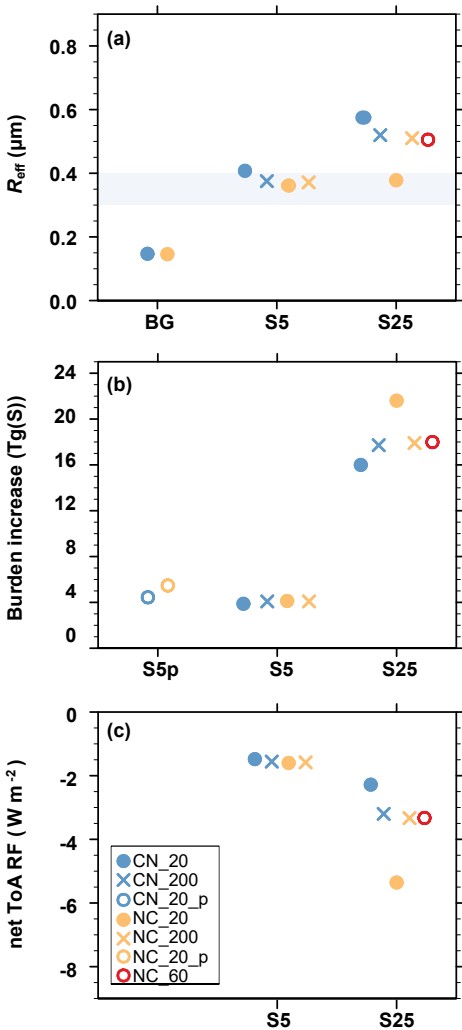

**Figure 3.** Effect of microphysical settings on the global averages of various calculated aerosol and radiative quantities. **(a)** Global mean effective radius (μm) at 55 hPa. **(b)** Global mean sulfuric acid aerosol burden increase (Tg(S)). **(c)** Global mean change in net top-of-atmosphere (shortwave + longwave) radiative forcing ($\mathrm{Wm^{-2}}$). Injection scenarios are: BG = background conditions (no injection); S5 = 5 Tg(S) yr$^{-1}$ regional $SO_2$ injection; S5p = 5 Tg(S) yr$^{-1}$ point $SO_2$ injection; and S25 = 25 Tg(S) yr$^{-1}$ regional $SO_2$ injection. Blue symbols: condensation first. Orange and red symbols: nucleation first. Open or filled circles: $N_{\mathrm{micro}} = 20$ (or 60). Crosses: $N_{\mathrm{micro}} = 200$. Light blue shading in **(a)**: optimal effective radii for scattering of solar radiation from Dykema et al. (2016) and Figure 4 in Weisenstein et al. (2022).

Figure 3b shows the impact of microphysical settings on the total (troposphere and stratosphere) aerosol burden increase in the intervention scenarios compared to background conditions. For background conditions, CN and CN with $N_{\mathrm{micro}} = 20$ show an almost identical aerosol burden (see also Table S1 in the Supplement). For the $SO_2$ injection scenarios, the original





setup CN_20 reveals the smallest aerosol burden. The largest aerosol burden is simulated by the simulation with CN_20 setting,
since this setup shifts the size distribution towards small particles, which have a longer stratospheric residence time. For the
S5 scenario the dependence on call sequence is small, but for S25 the simulated aerosol burdens differ by more than 30%
(Table S1). Despite this large spread in the simulated burden increase, SOCOL-AERv2 still falls between the CESM2 and
the MAECHAM5-HAM models, which showed for most of the simulated injection scenarios the largest and smallest burden
increase, respectively (Weisenstein et al., 2022, their Fig. 1).

Figure 3c displays globally averaged changes in the net top-of-atmosphere short- and longwave radiative forcing (RF) due to
$SO_2$ injections. Since SOCOL-AERv2 uses prescribed SST and SIC, the climate intervention runs remain in non-equilibrium
and the perturbation in radiative fluxes at TOA directly quantify the Effective RF (Forster et al., 2016). All S5 simulations
show a rather consistent RF change of around -1.5 $W\,m^{-2}$. For the S25 simulations, however, we find a large spread in RF,
ranging from -2.3 $W\,m^{-2}$ for the original microphysical setup (CN_20) to -5.4 $W\,m^{-2}$ for the simulation with reversed call
sequence (CN_20). As already mentioned in Weisenstein et al. (2022), the differences in RF between the various SOCOL-
AERv2 simulations, but also between different models, are mainly related to the respective burden increases (Fig. 3b). The
simulations with the largest burden increase also show the smallest $R_{eff}$, which efficiently scatters the incoming solar radiation
and enhances the negative RF.

As discussed in previous studies (Heckendorn et al., 2009; Kleinschmitt et al., 2018), the efficacy of the $SO_2$ injection,
i.e. the RF per Tg of sulfur injected annually, decreases with increasing injection rate, since the aerosol particles grow larger,
which increases sedimentation and decreases scattering efficiency. However, the model intercomparison by Weisenstein et al.
(2022) revealed that not only the radiative efficacy itself, but also its decrease with increasing injection rates is strongly
model dependent. For SOCOL in Fig. 3c, the radiative efficacy of the various S5 simulations ranges moderately between
-0.28 and -0.34 $W\,m^{-2}\,(Tg(S)\,yr^{-1})^{-1}$. For the S25 simulations, the simulations with highest and lowest efficacy differ
by more than a factor of 2. The applied microphysical improvements lead to a significantly higher radiative efficacy (0.09
$W\,m^{-2}\,(Tg(S)\,yr^{-1})^{-1}$ for S25_CN_60) compared to the default setup (0.13 $W\,m^{-2}\,(Tg(S)\,yr^{-1})^{-1}$ for S25_CN_20).

As SOCOL-AERv2 does not include an interactive ocean model, but prescribed SSTs, it is unfeasible to test the impact
of the call sequence on surface temperature anomalies. To overcome this limitation, we performed the G4 GeoMIP scenario
with the CN setup (S5p_CN_20) using the ESM SOCOLv4, a coupled model which shares the same exact aerosol module
as SOCOL-AERv2 (see methods). In this model, we use point injections, in keeping with the G4 protocol (see section 2.3).
The simulation shows an increase of 25% in stratospheric aerosol burden compared to the conventional S5p_CN_20 scenario
(see Fig. 3b, left). The corresponding global averaged surface cooling is 0.65 K and 1.02 K for S5p_CN_20 and S5p_CN_20,
respectively, which is an increase of 57%, whereas no significant differences in global stratospheric aerosol burden and RF
were found among regional S5 scenarios performed with SOCOL-AERv2 (see Fig. 3b, middle). This underlines the sensitivity
of our results to the chosen injection scenario (point vs. regional) as well as to the model resolution (section 2.2). Both the
model resolution and the injection scenario can lead to locally very different $H_2SO_4$ supersaturation.





## 3.3 Influence of settings on meridional distributions of aerosol burden, radiative forcing, and ozone

Figure 4a,b show the influence of microphysical settings on the modeled latitudinal variation of the sulfate aerosol column burden (stratosphere plus troposphere) for the climate intervention scenarios simulated with SOCOL-AERv2 (S5 and S25). In contrast, background simulations (not shown) have almost no dependence on the call sequence (see Table S1 in the Supplement). The $SO_2$ injection scenarios show similar latitudinal patterns, with aerosol column burdens peaking over the inner tropics, confined by the tropical leaky pipe. After overcoming the subtropical jet, the burden again maximizes around $45°N/S$ in the stratospheric surf zone, whereas the polar regions are isolated by the polar jets. As discussed before (Fig. 3b), the original setting CN_20 results in the smallest aerosol burden, whereas CN_20 with "nucleation first" shifts the size distribution towards smaller particles with less gravitational settling (see also Table S1).

The latitudinal variations of the radiative forcing (RF) in Fig. 4c,d show the mirror image of the burden in Fig. 4a,b, with reduced irradiance at high aerosol loading, and illustrate the direct radiative effects of the aerosol. However, in contrast to the smooth distributions of aerosol loading, RF exhibits a much higher degree of small fluctuations due to tropospheric cloud variability. The latitudinal variations in RF are very similar for all S5 simulations and the S25 simulations also show a consistent geographic pattern. The negative RF covers more than 80% of the globe, with the exception of the polar caps where absorption of outwelling infrared radiation by the aerosol predominates and the RF becomes positive. The differences between the individual simulations become largest in the tropics, reflecting the sensitivity of the aerosol loading to the microphysical setup.

Figure 4e,f shows the impact of the simulated $SO_2$ injections on zonally averaged total column ozone as difference to the reference simulation BG_CN_20. As already discussed by Weisenstein et al. (2022), the $SO_2$ injections lead to a massive reduction of the ozone column. This is caused by accelerated $ClO_x$-induced and $HO_x$-induced ozone destruction cycles, which in turn are due to heterogeneous $N_2O_5$ hydrolysis on the aerosol particles (leaving less $NO_2$ required for $ClO_x$ and $HO_x$ deactivation). The $N_2O_5$ hydrolysis rate is proportional to the SAD (see next section and Figs. S2 and S3 in the Supplement). Both injection scenarios, S5 and S25, show similar patterns with the most pronounced changes in mid- to high latitudes. In the polar regions, the ozone loss is mainly caused by enhanced heterogeneous $ClONO_2$ activation on the additional aerosol SAD. Furthermore, in agreement with the CESM2 model, SOCOL-AER simulates a decrease of the ozone column in the tropics, where the accelerated Brewer-Dobson circulation leaves less time for ozone formation by molecular oxygen photolysis. In the tropics, the presented microphysical modifications do not show any significant impact on the simulated ozone decrease (Fig. 4e and f), despite clear differences in the simulated SAD for the same sulfur injection (Figs. S2 and S3). This result indicates that above a certain threshold a further SAD increase does not affect the $NO_x$ cycle and its coupling to the $ClO_x$ and $HO_x$ cycles anymore. The fact that the S25 simulations result in a more pronounced total column ozone change than the S5 simulations is related to a more pronounced strengthening of the Brewer-Dobson circulation, which reduces the time for chemical ozone formation, and the increased stratospheric $H_2O$ entry, which enhances ozone loss by the $HO_x$ cycle (Tilmes et al., 2018).

In mid-to high latitudes both injection scenarios, S5 and S25, reveal substantial differences in the total ozone loss simulated, depending on the microphysical settings used in the simulations. For the S5 simulations (Fig. 4e), the total ozone losses over



Antarctica range between 24 and 30 DU. For the Northern Hemisphere, the spread in simulated polar ozone losses is with 6 to 24 DU even larger. For the S25 simulations (Fig. 4e and f), the simulated polar ozone loss range between 60 and 80 DU over the Southern Hemisphere, and between 35 and 60 DU over the Northern Hemisphere. It should be noted that the microphysical setup with the smallest ozone change in one hemisphere, does not necessarily also show the smallest ozone change on the other

325 hemisphere, which might be related to the dynamical variability.

It should be emphasized that the discussed changes in total column ozone caused by stratospheric $SO_2$ injections refer to stratospheric concentrations of ozone depleting substances and GHGs projected for the year 2040. With further decreasing stratospheric chlorine loadings in the future, the impact of the enhanced aerosol SAD under $SO_2$ injections on total column ozone might change as the coupling between the $ClO_x$ and $NO_x$ cycle becomes less important.

330 **3.4 Influence of settings on SAD and stratospheric temperature**

Climate intervention by stratospheric $SO_2$ emission yields an increase in aerosol surface area density (SAD), which enables heterogeneous chemical reactions such as $N_2O_5$ hydrolysis, but which is also an approximate measure of the extinction and, hence, the backscatter of shortwave radiation. Moreover, the aqueous $H_2SO_4$ aerosol absorbs outwelling longwave radiation, which increases the air temperature, with repercussions for stratospheric dynamics.

335 Both quantities, SAD and temperature, are also affected by the microphysical settings CN versus CN and $N_{micro}$. In brief, the CN setting with $N_{micro} = 200$ yields higher SAD than CN with $N_{micro} = 20$, roughly by 20%. This is due to the smaller particles with higher SAD and larger burden (see Figs. S2 and S3 in the Supplement). The larger burden, in turn, leads to more longwave radiative heating, which increases stratospheric temperatures. This is a marginal effect in the S5 scenario, but corresponds to about 1 K higher temperatures under S25 conditions (see Fig. S4 in the Supplement). A strongly temperature

340 dependent reaction such as $O_3 + O \rightarrow 2\,O_2$ changes by about 4% for $\Delta T = 1$ K, so that the impact of microphysical settings on ozone via SAD-changes is by far more important than the impact via $T$-changes. Also differences from dynamical feedbacks between the different settings are likely small since the absolute temperature increase from the S25 scenarios is up to 15 K and thus much larger.

**3.5 Influence of settings on modeling the eruption of Mt. Pinatubo**

345 So far, our study has highlighted the impacts of the microphysical settings for an extreme case involving climate intervention. Here, we expand this analysis, by evaluating the effects under conditions of volcanic eruptions on the 1991 eruption of Mt. Pinatubo by using the PIN_CN_20, PIN_CN_20 and the PIN_CN_200 settings (Table 1). We compared the evolution of the computed global stratospheric aerosol burden with SAGE and HIRS satellite data and the evolution of the computed mean effective particle radius with balloon measurements over Laramie (Wyoming, see Fig. 5). Details on the observational

350 data sets and their uncertainties as well as model and inter-model uncertainties can be found in Sukhodolov et al. (2018) and Quaglia et al. (2023). All model settings show a very similar peak in the stratospheric aerosol burden, but distinctly different declines during the years 1992/93. "Nucleation first" shifts the size distribution towards smaller particles, which have a longer stratospheric residence time. The slower decline is in better agreement with observations, although it should be mentioned that





**Figure 4.** Effect of microphysical settings on the zonal averages of various calculated aerosol, radiative and chemical quantities. Left column: regional $SO_2$ injections of 5 $Tg(S)\,yr^{-1}$. Right column: regional $SO_2$ injections of 25 $Tg(S)\,yr^{-1}$. **(a,b)** Sulfuric acid aerosol burden per square meter $(mg(S)\,m^{-2})$. **(c,d)** Zonal mean net top-of-atmosphere (shortwave + longwave) radiative forcing $(Wm^{-2})$. **(e,f)** Change in zonal average column ozone (Dobson units). Blue lines: simulations with condensation first; orange and red lines: nucleation first. Dashed lines: $N_{micro} = 20$ microphysical timesteps; solid orange and blue lines: $N_{micro} = 200$; solid red lines: $N_{micro} = 60$. All panels use the simulation BG_CN_20 as reference.



the agreement with observations strongly depends on the assumed $SO_2$ emissions profile (Quaglia et al., 2023). Regarding the

mean $R_{\text{eff}}$, PIN_CN_20 simulates smaller values than PIN_CN_20 for the first couple of months after the eruption and higher values later on, as PIN_CN_20 returns faster towards background conditions due to faster sedimentation of larger particles. Overall, the microphysical modifications do not overly influence the discrepancy between modeled and observed $R_{\text{eff}}$ (Fig. 5b).

However, other than under climate intervention conditions the evolution of the aerosol burden and $R_{\text{eff}}$ in the PIN_CN_200 scenario are much closer to PIN_CN_20 than to PIN_CN_20. The volcanic eruption is a point event in time and space, whereas

the geoengineering scenarios have continuous emissions across all longitude and 30°N and 30°S in latitude, which establish a steady-state situation. This leads to $H_2SO_4$ production rates, which locally are about 10'000-100'000 times larger in the Mt. Pinatubo case compared to S5 and S25. Since nucleation is exponentially dependant on the $H_2SO_4$ supersaturation this leads to erroneously large nucleation rates in the PIN_CN_20 scenario. Coagulation is not efficient enough to remove the large amount of nucleation mode particles in that scenario. When increasing $N_{\text{micro}}$ to 200 (PIN_CN_200) the burden and the $R_{\text{eff}}$ of the

plume evolve following the PIN_CN_20 scenario since local supersaturation are smaller now and coagulation can keep up with efficiently removing the nucleation mode particles.

Therefore, for volcanic eruptions, where $H_2SO_4$ supersaturations are locally much larger compared to climate engineering scenarios, the correct solution is much closer to CN_20, since otherwise nucleation would erroneously dominate over condensation. This is a good example how the very different distributions of $H_2SO_4$ supersaturation in space and time when

simulating volcanic eruptions and geoengineering scenarios lead to different challenges within aerosol microphysics schemes (Heckendorn et al., 2009; Vattioni et al., 2019).

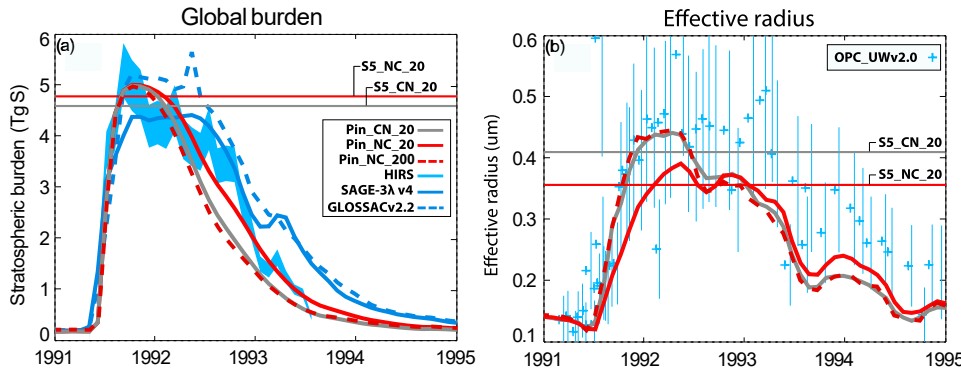

**Figure 5.** (a) Evolution of the simulated global stratospheric aerosol burden (Tg(S)) for PIN_CN_20 and PIN_CN_20 compared with HIRS- and SAGE-II-derived data (SAGE-3,4$\lambda$ and GLOSSACv2.2, Arfeuille et al., 2013; Thomason et al., 2018; Kovilakam et al., 2020). HIRS-derived total (troposphere and stratosphere) aerosol sulfur burden assumes 75% sulfuric acid by weight (Baran and Foot, 1994). Light blue shaded area: uncertainties of HIRS. (b) Effective particle radius (μm) averaged over the altitude range from 14 to 30 km compared to in-situ measurements taken at Laramie, Wyoming (OPC UWv2.0, Deshler et al., 2019). Thin blue whiskers reflect the measurement uncertainty (adapted from Quaglia et al., 2023). For comparison the steady-state values for S5_CN_20 and S5_CN_20 from this work are shown as thin horizontal red and gray lines in both panels.



## 4    Comparison with other work and conclusions

In this study, we have shown the importance of properly setting the length of the microphysical timestep and the call sequence
of nucleation and condensation for modeling the global stratospheric sulfuric acid aerosol under conditions of $SO_2$ injections
for climate engineering. In the aerosol-chemistry-climate model SOCOL-AERv2, the evolution of the $H_2SO_4$ concentration
in the gas-phase is determined by sequential operator splitting using a sub-stepping approach for aerosol microphysics with
a default timestep of 6 min, i.e. the $H_2SO_4$ gas-phase concentration is consecutively updated by $H_2SO_4$ production from
chemistry, condensation and nucleation. We found the following:

– Under stratospheric background conditions, the call sequence does not affect the model results, indicating that the default
380       number of microphysical sub-timesteps is sufficient to prevent the first called process from spuriously dominating the
size distribution.

– Under elevated $H_2SO_4$ supersaturations in the stratosphere the characteristic times for nucleation and condensation may
become shorter than the default microphysical timestep. In such cases, the competition between the two $H_2SO_4$ sinks
affects the simulated aerosol size distribution and the microphysical time step must be reduced.

– The default setting "condensation first" can massively underestimate the fraction of nucleation mode particles, whereas
"nucleation first" tends to underestimate the number of coarse mode particles. Tests of numerical convergence with
very short timesteps indicate that "nucleation first" yields smaller numerical errors for regional $SO_2$ injections, whereas
condensation first yields smaller numerical errors for the simulation of volcanic eruptions with locally and temporally
extremely high $H_2SO_4$ supersaturations.

– Despite significant shifts in simulated aerosol size distributions, the main response patterns of atmospheric chemistry
and climate to stratospheric $SO_2$ injections as simulated with SOCOL-AERv2 are robust to microphysical time inte-
gration adjustments, but the strength of the response can differ substantially in the case of high injection rates (such
as 25 $Tg(S)\,yr^{-1}$) or point injections (such as 5 $Tg(S)\,yr^{-1}$ point injections, S5p), which both lead to large $H_2SO_4$
supersaturations.

– The radiative forcing found for the 25 $Tg(S)\,yr^{-1}$ injection scenario varies by more than a factor of 2 between the
different microphysical settings. Nevertheless, this model-internal uncertainty in SOCOL-AERv2 is still smaller than
the scatter between the three GCMs with interactive aerosol microphysics – CESM2-WACCM6, MAECHAM5-HAM,
and SOCOL-AERv2 – compared by Weisenstein et al. (2022) in strictly coordinated climate intervention modeling.

The first part of our conclusions confirms the study by Wan et al. (2013), who investigated different time integration methods
to solve the $H_2SO_4$ continuity equation using two versions of the ECHAM-HAM model: HAM1 with an Euler forward scheme
with sequential operator splitting similar to SOCOL-AERv2, but without microphysical substeps; HAM2 with a two-step time
integration scheme implemented by Kokkola et al. (2009). They identified sequential operator splitting with too long timesteps
as major source of numerical error in HAM1, and proposed simultaneous processing of condensation and nucleation to better



represent the competition between both processes. The microphysical sub-stepping technique as applied in SOCOL-AERv2
improves the results of the operator splitting approach, but requires a sufficiently large number of substeps. Instead of using a
fixed number of substeps, a dynamical timestep adjustment could be beneficial, but we have not tested this here.

The importance of aerosol microphysics and the competition between nucleation and condensation on the simulated aerosol
size distribution and the radiative efficiency of stratospheric sulfur injections was also shown by Laakso et al. (2022), who in-
vestigated different injection strategies using the ECHAM-HAMMOZ model with two different aerosol schemes, the sectional
SALSA scheme as well as the modal M7 scheme. SALSA describes the aerosol size distribution in 10 size bins, while M7
uses 7 lognormal modes. The authors found that nucleation of new particles dominates over condensational particle growth in
the sectional SALSA scheme, while the opposite is the case in the modal M7 module. In addition, the use of lognormal modes
results in a minimum in the particle size distribution in the optimal size range for solar scattering and restricts the growth of
particles in accumulation mode, tending to underestimate gravitational settling. These differences resulted in smaller particles
in SALSA and, therefore, a higher radiative forcing. For an injection scenario of $20\,\mathrm{Tg(S)\,yr^{-1}}$, SALSA revealed a global net
ToA radiative forcing of around $8\,\mathrm{W\,m^{-2}}$, M7 resulted in $3\,\mathrm{W\,m^{-2}}$. This spread is even larger than what we found for the
S25 simulations S25_CN_20 and S25_CN_20. Laakso et al. (2022) further investigated the impact of the competition between
nucleation and condensation by performing simulations with the nucleation being switched off in both aerosol modules by
emitting 25% of the sulfur directly as $3\,\mathrm{nm}$ particles. The results of these sensitivity studies indicate that the different treat-
ment of nucleation and condensation explains the differences in radiative forcing between SALSA and M7 only partly: The
difference in radiative forcing was reduced from $5\,\mathrm{W\,m^{-2}}$ to about $3\,\mathrm{W\,m^{-2}}$.

Apart from time integration or representation of the aerosol size distribution, numerical parameterizations of individual
processes are another source of uncertainty. The binary-homogeneous nucleation scheme by Vehkamäki et al. (2002), for
example, is widely used in models, including SOCOL-AERv2 or the above mentioned aerosol schemes SALSA and M7.
The latter two include an extension of the scheme for high sulfate concentrations implemented by Kokkola et al. (2009),
using the collision rate as maximum possible nucleation rate. In a very recent study, Yu et al. (2023) evaluated simulated
nucleation rates in the lowermost stratosphere by CLOUD laboratory measurements under stratospheric temperatures. They
found that the Vehkamäki scheme overestimates observed nucleation rates by 3 to 4 orders of magnitude. As the particle size
distribution is not only determined by nucleation, but also by particle growth through condensation and coagulation, Yu et al.
(2023) compared the simulated size distributions to in-situ measurements of the particle number densities down to a diameter
of $3\,\mathrm{nm}$ obtained during the NASA Atmospheric Tomography Mission (ATom) between 2016 and 2018. In the size range
between 3 to $10\,\mathrm{nm}$, the number densities simulated with the GEOS-Chem model using the Vehkamäki et al. scheme were
1-2 orders of magnitude higher than observed. The same holds true for SOCOL-AERv2: under background conditions in the
southern hemisphere lowermost stratosphere (70 °S, $12\,\mathrm{km}$ altitude), modeled number densities for particles smaller than
$10\,\mathrm{nm}$ in diameter range between $10^3$ and $10^4\,\mathrm{cm^{-3}}$, while the ATom observations indicate values between slightly below
$10^1$ to $10^2\,\mathrm{cm^{-3}}$. Using the kinetic scheme for ion-mediated and binary homogeneous nucleation (Yu et al., 2020) calculated
nucleation rates, but also the size distributions simulated by GEOS-Chem were closer to ATom. Furthermore, the results by Yu
et al. (2023) suggest that under low stratospheric background $H_2SO_4$ concentrations nucleation on ions, which is usually not





represented in global models, dominates over binary homogeneous $H_2SO_4$-$H_2O$ nucleation. However, the importance of binary
homogeneous nucleation is expected to increase under high $H_2SO_4$ concentrations. Unfortunately, CLOUD measurements of
nucleation rates refer to stratospheric background conditions only and do not cover strongly elevated $H_2SO_4$ concentrations
under $SO_2$ injection scenarios or after volcanic eruptions, but based on the findings of Yu et al. (2023) it may be that all
models using the Vehkamäki scheme overestimate the role of nucleation. This might explain the low bias in the simulated
mean effective radius compared to in-situ measurements following the eruption of Mt. Pinatubo. Furthermore, this might
have substantial repercussions on the simulated aerosol size distribution, aerosol burdens and radiative forcing under climate
intervention conditions, most likely resulting in a decreased efficiency of $SO_2$ injections.

This work adds to a series of recent publications that highlight the crucial role of aerosol microphysics for simulated aerosol
properties and modeled estimates of geoengineering effects on atmospheric chemistry and climate. Our results clearly demon-
strate that there is considerable uncertainty when numerical schemes like the aerosol microphysics in SOCOL-AERv2 are
applied under unprecedented conditions, such as stratospheric solar geoengineering with continuously large $SO_2$ emissions,
even if these models had been thoroughly evaluated and are well capable of reproducing observations under background or
moderately perturbed conditions like volcanic eruptions. It should be emphasized that our conclusions are mainly based on
simulations of regional $SO_2$ injections, which are supported by point injection scenarios and simulations of the 1991 Mt.
Pinatubo eruption. As the nucleation rate strongly depends on the gas-phase $H_2SO_4$ concentration, ambient temperatures and
relative humidities, the optimal number of microphysical (sub-)timesteps will depend on the assumed $SO_2$ injection rates, but
also on the injection scenario and region. Point injections of $SO_2$, for example result in very high, but locally confined $H_2SO_4$
supersaturations, potentially making the results more sensitive to the details of the microphysical approach. The intention of
this paper is to raise awareness within the (aerosol) modelling community for potential numerical problems within conventional
aerosol microphysics modules when applying them to unprecedented extreme conditions such as high $H_2SO_4$ supersaturations
from $SO_2$ injection for climate intervention.

While this study focused on the importance of a proper temporal resolution of aerosol microphysics, it did not address ef-
fects of spatial resolution. Properly resolving the various temporal and spatial scales, ranging from nanometers and seconds for
microphysical processes to kilometers and decades for global climate, remains a significant challenge for aerosol-chemistry-
climate models (Vattioni et al., 2019; Weisenstein et al., 2022). Continuous model development, such as embedded $SO_2$ emis-
sion plume modelling (Sun et al., 2022), is indispensable to close the spatial and temporal gap between aircraft emission
plumes and large-scale model grids, and to effectively reduce existing model uncertainties with respect to the effectiveness of
geoengineering by stratospheric sulfur injections. Furthermore, additional laboratory or small-scale field studies of aerosol for-
mation, growth and dispersion under various stratospheric conditions could also be beneficial to evaluate and improve existing
numerical models.



*Supplement.* The supplement related to this article is available online at https://doi.org/10.5194/amt-16-xxx-2023-supplement.

*Code and data availability.*   The original SOCOl-AERv2 code is available at https://doi.org/10.5281/zenodo.5733121 (Brodowsky et al., 2018). The simulation data using that model code, which does not account for the interpolation of $H_2SO_4$ production within the microphysical subloop are available at https://doi.org/10.3929/ethz-b-000610854 (Stenke et al., 2023). The modified source code of SOCOL-AERv2 handling the microphysical subloop by taking into account the interpolation of $H_2SO_4$ production within the

microphysical subloop as well as the data from these simulations are available at http://hdl.handle.net/20.500.11850/622193 (Vattioni et al., 2023).

*Author contributions.*   SV proposed the study and implemented the modifications to the microphysical scheme. SV and AS performed the geoengineering simulations with SOCOL-AERv2 and analysed the model results. EW performed and analyzed the simulations with SOCOLv4. TS conducted and analyzed the Pinatubo simulations. Significant scientific guidance on the overall project was provided by TP

and BL. AS drafted a first version of the manuscript. All co-authors contributed to the discussion and the text.

*Competing interests.*   At least one of the (co-)authors is a member of the editorial board of Geoscientific Model Development. Other than this, the authors declare that they have no conflict of interest.

*Acknowledgements.*   Our special thanks go to Debra Weisenstein for discussions about the original AER code. We also thank Ilaria Quaglia for providing the processed OPC data. Support for Gabriel Chiodo and Andrea Stenke was provided by the Swiss Science Foundation

within the Ambizione grant no. PZ00P2_180043. Support for Sandro Vattioni was provided by the ETH Research grant no. ETH-1719-2 as well as by the Harvard Geoengineering Research Program. Timofei Sukhodolov acknowledges the support from the Swiss National Science Foundation (grant no. 200020-182239) and the Karbacher Fonds, Graubünden, Switzerland.





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
