# Peer review of "Importance of microphysical settings for climate forcing by stratospheric SO2 injections as modelled by SOCOL-AERv2"

_EGUsphere, 2023_

## Author Comment (AC1)

Dear Dr. Boucher,

Thank you very much for your comment, which we very appreciate. We address all your points raised in blue color below.

Best,

Andrea & Sandro & Co-Authors

The authors are right that the numerical aspects of the aerosol microphysical scheme should not be overlooked. In the S3A model (Kleinschmitt et al., 2017), we opted for an adaptive sub-timestepping approach as a compromise between accuracy and computation cost (see section 2.2.5 of the reference below for a full description). Here is an extract of our study without the equation:

"As both processes, nucleation and condensation, consume $H_2SO_4$ vapour while having very different effects on the particle size distribution, the competition between the two processes has to be handled carefully in a numerical model. Furthermore, this has to be done at an affordable numerical cost, as we aim to perform long global simulations. We address this in the S3A module using an adaptive sub-timestepping. After computing the $H_2SO_4$ fluxes due to nucleation and condensation in kg $H_2SO_4$ $s_{-1}$ from the initial $H_2SO_4$ mixing ratio, a sub-timestep, $\Delta t_1$, is computed such that the sum of both the nucleation and condensation fluxes consumes no more than 25 % of the available ambient $H2SO4$ vapour... This sub-timestepping procedure is repeated up to four times ... The fourth and final sub-timestep is chosen so that the sum of all sub-timesteps is equal to one timestep of the model atmospheric physics. This joint treatment of nucleation and condensation is imperfect, but it has the advantage of being much more computationally efficient than the usual solutions consisting of taking very short timesteps and much simpler than a simultaneous solving of nucleation and coagulation. The number of sub-timesteps could be increased for increased numerical accuracy; however, a number of four sub-timesteps was considered to be sufficient."

You may want to benchmark this approach (using different numbers of sub-timesteps) against yours in terms of accuracy and computational cost.

Reference

Kleinschmitt, C., Boucher, O., Bekki, S., Lott, F., and Platt, U.: The Sectional Stratospheric Sulfate Aerosol module (S3A-v1) within the LMDZ general circulation model: description and evaluation against stratospheric aerosol observations, Geosci. Model Dev., 10, 3359–3378, https://doi.org/10.5194/gmd-10-3359-2017, 2017.

Thank you very much for your comment and for pointing to Kleinschmitt et al. (2017). We had a look at that paper. We think that the method which is presented there is subject to similar problems described in this paper when exposed to continuously large $H_2SO_4$-supersaturation as they appear in a continuous $SO_2$ injection scenario. Same as our model, S3A works fine under background conditions and for the representation of volcanic eruptions. SOCOL-AER is also able to reproduce background sulfuric acid aerosol concentrations in the stratosphere under background conditions (e.g., Feinberg et al., 2019) as well as under conditions of volcanic eruptions (e.g., Sukhodolov et al., 2018, Quaglia et al., 2023). However, when exposed to continuously large $H_2SO_4$-supersaturations under conditions of continuous $SO_2$ injections the numerical solution of the semi-implicit scheme gets numerically unstable.

From the description in Kleinschmitt et al., 2017, it is not clear which $H_2SO_4$-supersaturation is used to calculate the nucleation rate:

"After computing the H2SO4 fluxes due to nucleation and condensation in kg H2SO4 s−1 from the initial H2SO4 mixing ratio, a sub-timestep, is computed such that the sum of both the nucleation and condensation fluxes consumes no more than 25 % of the available ambient H2SO4 vapour."

If understood correctly, Kleinschmitt et al. (2017) start with a large H2SO4-supersaturation resulting from the H2SO4-production of the previous 30 min dynamical timestep (i.e. $H2SO4_0$ in Kleinschmitt et al., 2017), which needs to be balanced by condensation and nucleation. However, using this initially very large H2SO4 supersaturation for the calculation of nucleation rates leads to significant overestimation of nucleation mass fluxes at large H2SO4 supersaturations. Please have a look at chapter S2 in the supplement of this manuscript. We show that it is important to properly distribute the H2SO4 production over the microphysical sub-loops as well. Just continuously updating the initially large $H2SO4_0$ concentrations after each sub-loop resulted in non-physical features in the resulting aerosol size distribution under high H2SO4 supersaturations.

Spitting the 30-min-time step into 4 parts proportional to 25% of the total H2SO4 nucleation and condensation mass flux probably results in timstep lents of t1<t2<t3<t4, where the first one is the shortest, due to higher supersaturations in the beginning and thus stronger condensation and especially nucleation rates. However, making the sub-loops proportional to mass flux is probably also subject to significant biases under larger H2SO4 supersaturations, since the nucleation flux is only a tiny fraction (about 1%) of the condensation flux when applying a very short microphysical time step (see our Table S1 in the supplementary material). Having a too large microphysical timestep in combination with larger H2SO4 supersaturations results in a strong overestimation of the nucleation rate.

The parameterisation presented in Kleinschmitt et al., 2017 might work for small H2SO4 supersaturations or short perturbations (e.g. Volcanic eruptions), but we suspect that the solution presented is subject to similar problems as we present in our manuscript. Maybe it might make sense to sensitivity test your model to conditions of continuously high H2SO4 supersaturations (i.e. SAI conditions).

Andrea & Sandro & Co-Authors

References:

Feinberg, A., T. Sukhodolov, B. P. Luo, E. Rozanov, L. H. E. Winkel, T. Peter, and A. Stenke (2019): Improved tropospheric and stratospheric sulfur cycle in the aerosol-chemistry climate model SOCOL-AERv2, Geosci. Model Dev., DOI:10.5194/gmd-2019-138.

Kleinschmitt, C., Boucher, O., Bekki, S., Lott, F., and Platt, U.: The Sectional Stratospheric Sulfate Aerosol module (S3A-v1) within the LMDZ general circulation model: description and evaluation against stratospheric aerosol observations, Geosci. Model Dev., 10, 3359–3378, https://doi.org/10.5194/gmd-10-3359-2017, 2017.

Quaglia, I., Timmreck, C., Niemeier, U., Visioni, D., Pitari, G., Brodowsky, C., Brühl, C., Dhomse, S. S., Franke, H., Laakso, A., Mann, G. W., Rozanov, E., and Sukhodolov, T.: Interactive stratospheric aerosol models' response to different amounts and altitudes of SO2 injection during the 1991 Pinatubo eruption, Atmospheric Chemistry and Physics, 23, 921–948, https://doi.org/10.5194/acp-23-921-2023, 2023.

Sukhodolov, T., Sheng, J.-X., Feinberg, A., Luo, B.-P., Peter, T., Revell, L., Stenke, A., Weisenstein, D. K., and Rozanov, E.: Stratospheric aerosol evolution after Pinatubo simulated with a coupled size-resolved aerosol–chemistry–climate model, SOCOL-AERv1.0, Geosci. Model Dev., 11, 2633–2647, https://doi.org/10.5194/gmd-11-2633-2018, 2018.

---

## Author Comment (AC2)

Dear Professor Visioni,

Thank you very much for your review, which helped us improve the manuscript. We appreciate the time you invested and address all your comments in blue color below.

Best,

Andrea & Sandro & Co-Authors

This is a rather important paper, neatly discussing the assumption behind SOCOL microphysical scheme and how changes in the assumptions that are valid in background conditions need to be re-assessed for simulations of SAI. The addition of the Pinatubo simulations and their discussion in light of the SAI finding is very nicely done. The manuscript is perfect for GMD and is exceptionally well written, so I think it should be promptly published. I have a just a few minor comments below.

Great abstract. I would suggest using terms now more widely used, like Solar Radiation *Modification* and just SRM (or SAI) instead of strat-SRM, which is confusing (in my opinion). You also never use the term "strat-SRM" in the manuscript, so a bit pointless.

Thanks for pointing this out. We stick to the term "Climate Intervention".

Line 112: "Neighboring size bins differ by molecule number doubling" this description is slightly confusin

We agree, since it is only a doubling for one of the two neighbors. We rewrote to: "Subsequent mass bins differ by molecule number doubling"

Line 182: Do they actually follow G4? G4 didn't explicitly mention how to inject (only indicating to do it just like in the models' simulations of Pinatubo), and used RCP4.5, while SSP5-8.5 is more recent than that. Are you talking about G6, which is based on SSP5-8.5 but goes to SSP2-4.5 surface temperatures, and prescribed injections at 10N-10S? If so, need to correct here. Right reference is more correctly Kravitz et al. (2015).

We corrected the scenario specifications: "These so-called "regional injections" are complemented by an example of a "point injection" performed with SOCOLv4 (see section 2.2) injecting 5 Tg(S) yr−1 in the form of SO2 at the same vertical extent but constrained to a region from 10°N to 10°S at the equator only emitting at the 0° meridian. These "point emission" scenarios with constant injection rates were motivated by the G4 GeoMIP experiment described in Kravitz et al. (2011). However, instead of RCP4.5 GHG and injections of 5 Mt SO2/yr as specified in Kravitz et al. (2011), we used SSP5-8.5 GHG and injected 5 Mt/yr S in the form of SO2, which is consistent with Wunderlin et al. (2024). The transient SSP5-8.5 boundary conditions allow us to explore the sensitivity of surface temperature to the call sequence in a fully coupled ESM."

Line 277-290: Again, some clarity needed in which scenario you used.

We corrected to: "To overcome this limitation, we performed a 5 Tg(S) yr−1 "point injection" scenario with the CN setup (S5p_CN_20, see section 2.3) using the ESM SOCOLv4, a coupled model which shares the same exact aerosol module as SOCOL-AERv2 (see methods)."

Line 345: why only the "extreme" one? You also consider a 5 Tg case which is not extreme by Pinatubo-like eruption standards.

We changed to : "So far, our study has highlighted the impacts of the microphysical settings for continuous injections in climate intervention scenarios."

Line 361: use exponential notation to avoid confusion here please (as you do elsewhere!).

Agree, we changed to exponential notation.

Line 369: a good example "of" how…

Thanks for spotting, this was corrected.

Line 450: old habits die hard… you use "solar geoengineering" here while saying it's a misleading term in line 34 and promising you'll use the term "climate intervention" in your work. I don't mind "geoengineering" as a term, but if you – and free to do so – then abide by the promise not to use it, so as to not to confuse the reader!

Thanks for pointing this out again. We are sticking to our promise now, and use the term "climate intervention".

---

## Author Comment (AC3)

Dear Dr. Niemeier,

Thank you very much for your review, which helped us improve the manuscript. We appreciate the time you invested and address all your comments in blue color below.

Best,

Andrea & Sandro & Co-Authors

The paper by Vattioni et al on the different handling of the operator splitting of nucleation and condensation and, sub-timestepping handles an important aspect of uncertainties in the evolution of sulfate aerosols in the stratosphere. The paper is well written and needs minor revisions.

My main comments are related to the last section. My recommendation is to discuss your work in more detail in relation to Wan et al (2013,2023), not just to the schemes in HAM 1 and HAM2, but also in relation to their method that solves production, condensation and nucleation simultaneously. You see, somehow, similar problems, and their article provides a solution. It may be difficult to upgrade your model, but you should discuss the conclusion that it would be better to solve multiple processes simultaneously. It may be good to add a figure of the nucleation rates as well.

We have addressed this in the section from 407 onward. We have added the following two sentences: "However, the best way forward would be to directly implement an implicit solver of H2SO4 production, condensation and nucleation into the next version of SOCOL as it is presented in Wan et al. (2013). This will likely generate more accurate results while avoiding the need for sub-loops, which are computationally more expensive. However, these implicit solvers have not yet been tested for numerical stability under conditions of continuously larger SO2 injections as they would occur in climate intervention scenarios."

We think the nucleation rates provided in Table S1 in the supplement are enough. An additional figure for nucleation rates only would be beyond the scope of the paper.

Can you give a recommendation how to proceed? What is the best option for simulations of SAI or volcanic eruptions? Will this also apply to other models?

An outlook is given starting from lines 462 onward. There is no best "option" for simulation of SAI or volcanic eruptions. The microphysical settings are very sensitive to the model resolution (horizontal, vertical and temporal) as well as to the specific injection/emission scenario applied (location and injection rate and time). It is unclear if the numerical instability under large H2SO4-supersaturations also applies to other models, however, it is definitely something other modeling groups should be aware of. This is discussed in the indicated paragraph. We have added the following sentence as a recommendation how to proceed:

"To increase confidence about different aerosol microphysics modules incorporated in the various aerosol-chemistry climate models we recommend conducting a model intercomparison study, which focuses on numerical stability of aerosol microphysics under conditions of large H2SO4 supersaturation."

Linee 7 and 10: 'of of'

Thanks for spotting, we corrected this.

Line 38: The aerosol scatter

We changed to "scatter"

Line 140: How do you handle other processes, e.g. sedimentation? A long timestep (2h) may reduce sedimentation artificially in case the aerosols sediment into the next gridbox only.

The simple updraft scheme used for sedimentation in the original 2-D AER code was replaced by the numerical scheme of Walcek et al. (2000) in Feinberg et al. (2019). This reduced numerical diffusion and improved mass conservation.

The sedimentation speed of a 320 nm 70wt%-$H_2SO_4$ aerosol is about 1 meter per 2 hours at 50 hPa in the stratosphere. For smaller particles it is even less. Therefore, calling the sedimentation routine every two hours is fine. We do not expect artificial reduction or increase of sedimentation.

2.2 Socolv4: Changing the resolution of the model changes transport (e.g. Niemeier et al, 2020). This should be kept in mind.

Yes, thank you for the comment. This is also one of the reasons why we took the "point injections" performed with SOCOLv4 into account in this paper. The microphysical configuration and the timestep applied do not only matter for large injection rates but also for injection scenarios with smaller injection rates combined with more confined injection locations.

Line 235 pp: There is a mismatch of the names. You write very often CN and CN. One should be NC.

Thanks for pointing this out. We corrected this.

Fig 3: You average between 30N to 30S in Fig 2. Fig 3 shows a global average. To see the differences between the simulations you may add a 30N to 30S average in Fig 3.

The average between 30N to 30S at 50 hPa was only chosen because this represents the injection region. Therefore, averaging over this region best represents the aerosol formation processes without biases through advection and sedimentation of aerosols. However, for quantities such as effective radius, aerosol burden and radiative forcing it makes more sense to look at global averages.

Line 301 - 303: This is not true for Fig 4c. CN_20 is more similar to CN_200 compared to the burden plot. Why?

We changed to: "The latitudinal variations of the burden in Fig. 4a,b are reflected in the changes in radiative forcing (RF) in Fig. 4c,d, with reduced irradiance at high aerosol loading, and illustrate the direct radiative effects of the aerosol. However, in contrast to the smooth distributions of aerosol loading, RF exhibits a much higher degree of small fluctuations due to tropospheric cloud variability." The term "mirrored" might have been over exaggerated. However, CN_20 is only more similar to CN_200 over the south polar region. In a global average CN_20 clearly results in less RF compared to the other scenarios (see Figure 2). The largest contribution to the global average results from the tropics where CN_20 results in the smallest RF.

Line 312: Less time for ozone formation or stronger meridional transport? The last might be quite important.

You are right, it is rather the stronger meridional transport. We corrected this.

Line 321-322: Where? 6 to 24 DU are the values for the hemisphere, not at high latitudes.

Thanks for pointing to this. The values in the text were wrong. We corrected them.

Section 3.5: Pinatubo is not a good analogue for SAI. The injection rate is much higher as are the SO2 concentrations. It might be of interest for you to compare with Wrana et al (2023). After the eruption of Ulawun, satellite data show a decrease in particle size. So nucleation after the eruption is important to get a good agreement between model and data.

We do not claim that Pinatubo is an analogue to SAI. Volcanic eruptions are not a good analogue to SAI in general independent of the injection amount. This is due to the continuous supply of H2SO4 in SAI scenarios, which is not the case in volcanic eruptions (e.g. Heckendorn et al. or 2009 or Vattioni et al., 2019). We only have chosen Mt. Pinatubo eruption for sensitivity testing to the microphysical settings because it is a large eruption and thus, the effects of the microphysical settings become most apparent.

Line 399pp: Wan et al 2013 offer a solution of your problem: 'These errors can be significantly reduced by employing solvers that handle production, condensation and nucleation at the same time.' You should discuss this - employing in the model might be an even better solution. You may have a look for Wan et al (2023) as well (https://doi.org/10.48550/arXiv.2306.05377).

Indeed employing solvers which handle production, condensation and nucleation in parallel as described in Wan et al. (2013) would be optimal. We are thinking of implementing this into the next version of SOCOL. However, this would be beyond the scope of this project. We think the findings of Wan et al. 2013 are already discussed in enough detail in the paragraph you are pointing to.

Line 412-414: This is not true in general, only for the specific setup of the modes in combination with the injection strategy used in Laakso et al (2022).

Thank you, we added a side note: "In addition, the use of M7 with lognormal modes results in a minimum in the particle size distribution in the optimal size range for solar scattering due to the accumulation mode reaching its largest size, which adds mass to the coarse mode in the injection scenario applied in Laakso et al. (2022). The resulting gap between the two modes tends to underestimate gravitational settling."

Line 415-416: spread: How is this related to this work. Aren't you comparing apples and pears here?

No, we do not compare apples with pears. We clearly state what we compare. It is important to highlight that different parameterizations of microphysical processes can result in substantial differences in resulting size distributions and thus, in resulting radiative forcing. Obviously, we can not directly compare the Lakso et al. 2022 study with our study due to different injection rates and injection scenarios. Therefore, we only point to the differences between models which apply different microphysical parameterizations. The differences in radiative forcing are even larger than the differences resulting from switching the calculation order and the radiative forcing in our model.

Line 424: As far as I understand Laakso (2022), SALSA does not use Vehkamäki. Nucleation in SALSA is much stronger than in HAM.

For SALSA Laakso et al. (2022) is pointing to Kakkola et al. (2018). However, there is no information about the type of parameterisation used in SALSA. In Kakkola et al. (2009)

state that SALSA uses the Vehkämäki scheme. Laakso et al. (2022) also points to Vignati et al. (2004) for M7, which states that M7 uses the Vehkämäki scheme.

Line 426: The collision rate is important for high SO2 concentrations. Otherwise the parameterizations of Vehkamäki (2002) might be not valid at all grid points. However, this is not well published. Määtänen et al (2018) is an upgrade and includes the collision rate. It might be worth to think about an implementation in your model.

Thank you for the comment. We will update the nucleation scheme to the one used in Määtänen et al. (2018) in the next version of SOCOL. See also answer to the following comment.

Line 443: Can you relate your results to Yu et al (2023)? You get very different answers with one nucleation parameterization, but different substepping etc. So, I wonder if this very general conclusion of Yu et al (2023) holds for your results. In Wrana et al (2023) we use Määtänen et al (2018). This parameterization reproduces the particle size after the Raikoke and Ulawun eruptions quite well. These small eruptions are much closer to SAI because they have a more similar eruption rate than the Pinatubo eruption. I recommend that you do a simulation, even if you do not want to include the results in this paper. You may gain more confidence, which is the better way to continue.

Yu et al. (2023) write: "In GC–APM, nucleation is calculated before condensation using a time-splitting technique. Therefore, no competition between nucleation and condensation for sulfuric acid vapor is considered. In most conditions, nucleation consumes only a very small fraction (<1 %) of sulfuric acid vapor in the air, and the time splitting does not affect the results. When the nucleation rate is high, a reduced time step for nucleation and growth is used to ensure that the fraction of sulfuric acid vapor consumed by nucleation each time step is small. The GC-APM uses a semi-implicit scheme to calculate sulfuric acid condensation together with sulfuric acid gas-phase production to ensure that the change of sulfuric acid vapor concentration is smooth."

It is unclear how our findings would affect the results reported by Yu et al. (2023). The solver in Yu et al. (2023) does not account for competition between nucleation and condensation and is semi-implicit together with a time step reduction for high H2SO4 supersaturations. The fact that nucleation consumes at most 1% of the sulfuric acid vapour is a good sign that the scheme is likely performing fine.

Indeed, the nucleation parameterization used in Määtänen et al. (2018) would likely be more appropriate to use in aerosol-chemistry climate models as it results in better agreement with observations. We will incorporate this in future versions of SOCOL. However, even though the local injection rate observed after smaller eruptions such as the Raikoke and Ulawun eruptions are more close to conditions of potential SAI scenarios, they are likely still not comparable to the aerosol size distributions resulting from continuous injections of SO2 in SAI scenarios (see Vattioni et al., 2019, or Heckendorn et al., 2009). However, we agree that we could learn from simulating these smaller eruptions and comparing them to observations and other models. We simply do not have the resources at the moment to do this analysis. We have added two sentences to the discussion section of the paper: "The reported weaknesses of the Vehkämäki scheme were addressed by Määttänen et al. (2018), who presented a new parameterisatio for sulfuric acid aerosol formation including homogeneous and ion-induced nucleation pathways validated by CLOUD laboratory measurements. The Määtäen et al. nucleation scheme which is reported to be valid for the whole range of atmospheric conditions including high stratospheric sulfuric acid concentrations during SO2 injection scenarios is recommended to be used in

aerosol-chemistry climate models instead of the Vehkamäki scheme (Määttänen et al., 2018)."

References

Määtänen, A., Merikanto, J., Henschel, H., Duplissy, J., Makkonen, R., Ortega, I. K., and Vehkamaki, H.: New Parameterizations for Neutral and Ion-Induced Sulfuric Acid-Water Particle Formation in Nucleation and Kinetic Regimes, J. Geophys. Res.-Atmos., 123, 1269–1296, https://doi.org/10.1002/2017JD027429, 2018.

Niemeier, U., Richter, J. H., and Tilmes, S.: Differing responses of the quasi-biennial oscillation to artificial SO2 injections in two global models, Atmos. Chem. Phys., 20, 8975–8987, doi.org/10.5194/acp-20-8975-2020, 2020.

Wrana, F., Niemeier, U., Thomason, L. W., Wallis, S., and von Savigny, C.: Stratospheric aerosol size reduction after volcanic eruptions, Atmos. Chem. Phys., 23, 9725–9743, https://doi.org/10.5194/acp-23-9725-2023, 2023.

References:

Walcek, C. J.: Minor flux adjustment near mixing ratio extremes for simplified yet highly accurate monotonic calculation of tracer advection, J. Geophys. Res., 105, 9335–9348, https://doi.org/10.1029/1999JD901142, 2000.

Feinberg, A., T. Sukhodolov, B. P. Luo, E. Rozanov, L. H. E. Winkel, T. Peter, and A. Stenke (2019): Improved tropospheric and stratospheric sulfur cycle in the aerosol-chemistry-climate model SOCOL-AERv2, Geosci. Model Dev., DOI:10.5194/gmd-2019-138.

Heckendorn, P., Weisenstein, D., Fueglistaler, S., Luo, B. P., Rozanov, E., Schraner, M., Thomason, L. W., and Peter, T.: The impact of geoengineering aerosols on stratospheric temperature and ozone, Environ. Res. Lett., 4, 045108, https://doi.org/10.1088/1748-9326/4/4/045108, 2009.

Vattioni, S., Weisenstein, D., Keith, D., Feinberg, A., Peter, T., and Stenke, A.: Exploring accumulation-mode H2SO4 versus SO2 630 stratospheric sulfate geoengineering in a sectional aerosol–chemistry–climate model, Atmos. Chem. Phys., 19, 4877–4897, https://doi.org/10.5194/acp-19-4877-2019, 2019.

Vignati, E., Wilson, J., and Stier, P.: M7: An efficient size resolved aerosol microphysics module for large scale aerosol transport, J. Geophys. Res.-Atmos., 109, D22202, https://doi.org/10.1029/2003JD004485, 2004.

Yu, F., Luo, G., Nair, A. A., Eastham, S., Williamson, C. J., Kupc, A., and Brock, C. A.: Particle number concentrations and size distributions in the stratosphere: implications of nucleation mechanisms and particle microphysics, Atmos. Chem. Phys., 23, 1863–1877, https://doi.org/10.5194/acp-23-1863-2023, 2023.

---

## Author Comment (AC5)

Dear Dr. Laakso,

Thank you very much for your review, which helped us improve the manuscript. We appreciate the time you invested and address all your comments in blue color below.

Best,

Andrea & Sandro & Co-Authors

The manuscript authored by Vattioni, Stenke, et al. examines the impact of sequential operator splitting and the number/length of time steps on the simulated outcomes of climate intervention through stratospheric sulfur injections. Authors say in the manuscript that "The intention of this paper is to raise awareness within the (aerosol) modelling community for potential numerical problems within conventional aerosol microphysics modules" and it does that very well. The key finding and message for the modeling community is that, even when simulating the same scenario with an identical model and identical lines of code, results can vary significantly (in this instance, ranging from a radiative forcing of -2.3 Wm-2 to -5.3 Wm-2) due to the order in which the two processes are calculated. The aspect of this "choice" is a detail which is quite often overlooked in studies related to stratospheric aerosol injection and model intercomparison studies. Furthermore, the manuscript is exceptionally well-written, making it straightforward to read and devoid of any major issues. Consequently, I strongly recommend the publication of this manuscript.

I have only some minor comments or corrections. First more general ones:

I would like to see some discussion from the authors about the role of the coagulation in all of this. It clearly had an impact in Pinatubo simulations (which was a nice point by the way!), but does it have a major impact in the case of stratospheric aerosol injections? In these simulations coagulation was included in the microphysical subloop. What do you think, would it have had a major impact if the number of steps would have been increased only for nucleation and condensation, but not coagulation? And just to be sure: I do not expect to see any additional simulation or analysis on this, but it would be just interesting to know if the authors would have any thoughts about this. Usually coagulation is a computationally heavy process (however probably not that much in SOCOL, where the coagulation kernel is not calculated inside the subloop) and thus not wanted to be calculated more than is needed.

This is a very good comment. We looked at this early on and also performed some sensitivity simulations on this with the S25 simulations, but then we decided to not highlight this in the paper. In these sensitivity tests we looked at the role of nucleation/condensation vs. coagulation by applying a microphysical sub-sub loop in the S25 simulations. This means, in addition to the microphysical sub-loop, we applied another sub-sub-lop for either coagulation only ("5_coag" in Figure 1 below) or for nucleation and condensation only ("10_nuc_con" and "2_nuc_con" in Figure 1) within the microphysical sub loop. We performed two simulations with a microphysical sub-step of 20 - one with 5 additional sub-sub-step for coagulation (i.e. 20 x nuc/cond and 20x5=100x coagulation) and the other with 10 additional sub-sub-steps for nucleation and condensation (i.e. 20x10=200 times nuc/cond and 20 times coagulation).

As expected, the additional sub-loops for coagulation only (5_coag) does not significantly reduce the nucleation mode particles. This is due to the very large H2SO4 supersaturation resulting from only having 20 sub-loops for nucleation and condensation, which results in very large nucleation mode particle concentrations. However, in the accumulation mode the particle concentrations are clearly reduced, due to efficient coagulation.

Applying 10 additional microphysical sub-sub-steps for nucleation and condensation only (10_nuc_con), results in a total of 20x10=200 sub-steps for nucleation and condensation and only 20 for coagulation. Here, we observe a clear reduction of the nucleation mode particle concentrations, which is due to the lower supersaturation when having a smaller timestep for nucleation and condensation. However, due to the inefficient coagulation (20 sub-steps only) the concentration of the accumulation mode particles is too large compared to NC_200. This also explains why the nucleation mode concentration is even lower than in the case for NC_200: When we have large accumulation mode particle concentrations (immobile targets), coagulation with the tiny nucleation mode particles (very mobile projectiles) is more efficient.

To conclude, only increasing the timesteps of nucleation/condensation or coagulation does not result in satisfying results. Therefore, both, the timesteps of nucleation/condensation and coagulation must be reduced. In additional simulations we found that a microphysical timestep of 40 combined with an additional sub-loop for nucleation/condensation, resulting in 80 (i.e., 40 x 2) sub-steps for nucleation/condensation and 40 sub-steps for coagulation would be enough to be very close to the S25_NC_200 solution. As expected, nucleation/condensation required a smaller timestep since the timescale of coagulation is usually larger compared to the one of nucleation/condensation. Therefore, to save computer time, the number of sub-loops for coagulation need not be increased as much as the one for nucleation/condensation, but it should nevertheless be sufficiently small.

However, the "optimal" amount of "sub-steps" and "sub-sub-steps" is highly dependent on the scenario applied (spatial confinement of the injections, pulsed or continuous injections, injection location and time) and is for example very different for the Mt. Pinatubo case. Therefore, we decided to not show these results in the main manuscript, since this could result in the misleading conclusion, that there is an "universal optimal solution".

We added the following sentence to line 234/235 of the new manuscript: "Since coagulation has the largest timescale and is computationally the most expensive process within the microphysical sub-loop, we also tested a scenario with 80 sub-steps for nucleation and condensation, but only 40 for coagulation (not shown)."

[Figure]

**Figure 1:** Sensitivity simulations of the S25 scenario for resulting size distribution (a) and "5th moment" (b) for an additional microphysical sub-sub-loop for nucleation/condensation only (nuc_con) and for coagulation only (coag). The burden in the legend indicates the total aerosol burden increase of each simulation compared to the reference run. The simulations shown were run with the same boundary conditions as the simulations in the main manuscript, with the only difference that here a one-year average of the year 2067 of a transient simulation is shown. This explains small deviations from the simulations shown in the main paper.

Authors also mentioned about the sensitivity of results to the chosen injection scenario, but this could be discussed little more in respect to take home messages of this study. What I mean is that e.g. increasing the number of timesteps, or calling nucleation routine first, had a large impact on S25 simulations but rather small for S5 simulations. Thus someone might think that this issue of calculating microphysics does not matter for 5 Tg(S)/yr injections. However, in this study the injections were done in a quite wide area/band (30 N - 30 S latitudes) and I assume there would have been a larger difference if the injections would have been done to a narrower band. This was actually shown more or less by ESM SOCOLc4 simulations for S5p.

Thank you for pointing this out. The conclusions could indeed have given the wrong impression that this issue only applies to large injection rates, which is wong as we show with the S5p simulations. We included the "S5p" results with SOCOLv4 in the last bullet point of the conclusions. And also, we put more focus to this aspect in the second last paragraph of the conclusions now:

"It should be emphasized that our conclusions are mainly based on simulations of "regional" $SO_2$ injections, which are supported by "point injection" scenarios and simulations of the 1991 Mt. Pinatubo eruption. As the nucleation rate strongly depends on the gas-phase $H_2SO_4$ concentration, ambient temperatures and relative humidities, the optimal number of microphysical (sub-)timesteps will depend on the assumed $SO_2$ injection rates, but also on the injection scenario and spatial confinement of the injections. "Point" injections of $SO_2$, for example result in very high, but locally confined $H_2SO_4$ supersaturations, which makes the results more sensitive to the details of the microphysical approach. This effect is shown with the "point injection" scenarios (S5p), which are much more sensitive to the microphysical settings compared to the "regional injection" scenarios (S5, see Fig. 3b)."

Some specific comments:

Some lines in the text there are "CN" while it was clearly referring to "nucleation first"/NC. Please check these (or neglect me if I have understood something wrong):

P9 L209 "nucleation first" (CN) -> "nucleation first" (NC)

P9 L214 "of the CN and CN simulations" -> "of the CN and NC simulations"

P10 L239 "CN and CN" -> "CN and NC"

P10 L245 "(CN, " -> "(NC, "

P11 L251 "CN and CN" ->"CN and NC"

P12 L253 "CN_20 setting" ->"NC_20 setting"

P12 L265 "sequence (CN_20)" ->

P12 L282, P13 L294,

P14 L335, L247,

P16 L355, L361, 363-365

Fig5 text

P18 L416

Thanks for pointing to the NC and CN chaos. We corrected all wrong abbreviations and updated the manuscript accordingly.

I noticed that this is something that Daniele (RC1) commented on already but I will mention it anyway: In the introduction it is said that "climate intervention" is used instead of "geoengineering", but still "geoengineering" is used in a couple of lines in the text.

Thank you for pointing to this. We made the terms consistent and use "climate intervention"

P1 L15 "25 MT/yr" -> "25 Tg(S)/yr" and "timesetp" -> "timestep"

We corrected this and made it consistent.

P5 L137 "H_2O_2" -> "H_2O"?

We corrected the typo to "$H_2SO_4$".

P6 L159 Maybe you could add that one major difference between SOCOL model versions is also the atmospheric model (ECHAM5 / ECHAM6). If I am not totally wrong.

ECHAM6 is part of MPI-ESM1.2, but we included a note that SOCOLv4 is based on ECHAM6 to make it clear.

P7 L177-184 I have to admit that I don't always completely remember all GeoMIP scenarios but here it is said that the simulated point scenarios are following the G4 GeoMIP scenario of Kravitz 2011. However G4 is based on RCP4.5 and 5 Tg $SO_2$/yr is injected (= 2.5 Tg(S)/yr). I assume you meant to refer to some other GeoMIP experiment? I was also thinking that "point" might be slightly misleading as the injections are done along several grid points along the meridian and thus it is different from in Pinatubo simulation. But there is no perfect way to name these and I do not have a better suggestion for the name.

We specified as follows:

These "point emission" scenarios followed the G4 GeoMIP scenario described in Kravitz et al. (2011). However, instead of RCP4.5 GHG and injections of 5 Mt $SO_2$ /yr as specified in Kravitz et al. (2011), we used SSP5-8.5 GHG and injected 5 Mt/yr S in the form of $SO_2$, which is consistent with Wunderlin et al. (2024). The transient SSP5-8.5 boundary conditions allow us to explore the sensitivity of surface temperature to the call sequence in a fully coupled ESM.

P 7 L 190-193 You could describe the Pinatubo experiment also here in text as it is done in Table 1. I mean how much $SO_2$ is injected, which altitude and which ISAMIP experiment you are referring to. You could also mention why you chose "low-shallow-injection scenario".

Thanks for pointing to this. We specified:

"In the absence of observational data of the stratospheric aerosol layer under climate intervention conditions, we also tested the effect of different microphysical settings in the modeling of the 1991 Mt. Pinatubo eruption. The 1991 Mt. Pinatubo eruption was specified

as 5 Tg S emitted in the form of $SO_2$ at 21-23 km altitude (2 model levels) above the Mt. Pinatubo geographical location (i.e., two model grid boxes) during one day. This set up corresponds to the HErSEA_Pin_El_Ism scenario proposed by the Interactive Stratospheric Aerosol Model Intercomparision Project (ISA-MIP, Timmreck et al., 2018), which has been shown to have better agreement with observations for some variables in (Quaglia et al., 2023) compared to scenarios with larger emission amounts and different emission altitudes"

P10, Fig. 2 Please check that (a)-(f) in the text (description of the figure) corresponds to the ones in the figure.

Thanks for spotting, I corrected this.

P11, Fig. 11. I recommend using some other color than light blue for optimal effective radii or at least make it darker. I did not see it when I printed the manuscript and it is not very clear in the pdf.

We changed the color to light green and made it less transparent.

P11, Fig. 11 or P12 L260, This probably would not need new figures and can be just mentioned in the text but it would be really interesting to see individual radiative forcing for shortwave and longwave radiation separately. I would be expecting that the difference in LW radiative forcing between simulations is relatively small. By the way, if radiative forcing is calculated as difference of radiative fluxes between perturbed and control/background scenarios, and not by e.g. double radiation call with and without aerosols, you might see some change in LW radiative forcing due to the land temperature adjustment which might be relatively large in case of 25 Tg(S)/yr injections. Of course this is something that does not affect your conclusions, but good to consider.

Yes, you are right (see Figure 2 below). The LW RF is very similar between simulations. There is also no significant difference between the S5 and the S25 simulations (see Figure 2 below) and thus, no significant effect from the land adjustment in the LW.

[Figure]

**Figure 2:** This Figure shows the SW and LW RF efficiency (RF/Mt injected material). The LW efficiency are the upper points and crosses (positive numbers) and the SW efficiency are the lower points and crosses (negative numbers).

P12 L284 This is more just a comment, but it is really interesting that the difference in temperature response between CN and NC scenarios is quite large here. As I mentioned

above, I am expecting the difference to be caused by the fact that the latitude band is narrower than for SOCOL-AERv2 simulations. In Laakso et al. 2022 point injection led to more similar response between SALSA (prefers nucleation) and M7 (prefers condensation), but there sulfur was injected to the one model grid point which probably gave nucleation more suitable conditions to fight against condensation in M7 simulations.

Yes, this is also our suspicion. The "point" injections (S5p) result in about 1000 times more confined injection regions (10°N to 10°S at one latitudinal band instead of 30°S to 30°N at all longitudes) and thus larger local $H_2SO_4$ concentrations. Nucleation increases exponentially to larger $H_2SO_4$ supersaturations and is thus much stronger in regionally more confined injection scenarios. As mentioned in your second major commend above, we have now highlighted this artefact more in the revised manuscript. However, it should also be kept in mind that also the model resolution is different between the models, which can also result in different results.

P13 L317 I am not sure if I understood how nudging of winds was done. I assume it was not fully nudged if there are changes in atmospheric circulation?

SOCOL-AER simulates the wind fields interactively. However, the model version with 39 vertical pressure levels does not generate a QBO. Therefore, the QBO is artificially imposed by a linear relaxation of the simulated zonal winds in the equatorial stratosphere to observed wind profiles over Singapore perpetually repeating the years 1999 and 2000 (due to a time slice set up in our main case). Thus, SOCOL-AER simulates its own wind fields, but the QBO is nudged towards observed wind fields. See Stenke et al., 2013 for details.

P18 L414. Actually, in M7 simulations of Laakso et al. 2022 growth of the particles in accumulation mode was not restricted, but the size of the accumulation mode was. I mean that accumulation mode was in its maximum size, and when particles in accumulation mode grew up (by condensation or coagulation), they were transferred to coarse mode. This created a gap between these two modes.

Thanks for pointing this out. I changed to "In addition, the use of lognormal modes results in a minimum in the particle size distribution in the optimal size range for solar scattering due to the accumulation mode reaching its largest size, which adds mass to the coarse mode. The resulting gap between the two modes tends to underestimate gravitational settling."

References:

Kravitz, B., Robock, A., Boucher, O., Schmidt, H., Taylor, K. E., Stenchikov, G., and Schulz, M.: The Geoengineering Model Intercomparison Project (GeoMIP), Atmospheric Science Letters, 12, 162–167, https://doi.org/10.1002/asl.316, 2011

Stenke, A., Schraner, M., Rozanov, E., Egorova, T., Luo, B., and Peter, T.: The SOCOL version 3.0 chemistry–climate model: description, evaluation, and implications from an advanced transport algorithm, Geosci. Model Dev., 6, 1407–1427, https://doi.org/10.5194/gmd-6-1407-2013, 2013.

Wunderlin, E., Chiodo, G., Sukhodolov, T., Vattioni, S., Visioni, D., and Tilmes, S.: Side Effects of Sulfur-Based Geoengineering Due To Absorptivity of Sulfate Aerosols, Geophysical Research Letters, 51, e2023GL107 285, https://doi.org/10.1029/2023GL107285, 2024

---

## Author Response (AR2)

Dear Dr. Graham Mann,

Thank you for your review and your suggestions. We addressed all of your comments and answer below in blue and followed your suggestions. We hope the manuscripts is now ready for publication. Thank you for handling this manuscript.

Best,

Andrea Stenke, Sandro Vattioni & Co-Authors

The three main reviewers (Visioni, Niemeier, Laakso) have each found the manuscript suitable for publication after minor revisions, and have graded the MS excellent or good in all 4 of the categories. Together also with a further review from a 4th scientist (Boucher), the comments together comprised quite a thorough review process.

The authors have replied to the 4 reviews, and I have been through these, and the author replies, and can confirm the authors having replied to each of the specified minor-revisions, and revised the manuscript accordingly.

The revised manuscript then has addressed the minor revisions raised by the 4 reviewers.

However, from checking the revised manuscript it's clear there were still a few further typo-level changes required, and I have listed these below as a final Topical Editor review.

These are very minor changes however, and once these 7 minor edits are addressed, the manuscript can then proceed to publication in GMD.

Topical-editor remaining typo-revisions
* * *
1) Abstract lines 1-2 – change of "Solar radiation management" to "Climate intervention"

I can see that 1 of the 4 reviewers suggested to change "solar radiation management", instead to "climate intervention". The authors have made that suggested change in line 1, and have deleted also the term SRM in 5 other places in the Abstract.

As a Topical Editor in this particular field, I was actually quite surprised this reviewer has the opinion the terminology strat-SRM can be "confusing". I checked for example the recent interactive stratospheric aerosol community intercomparison paper by Weisenstein et al. (2022), and see the Introduction sets the context of the article based primarily on the SRM acronym there.

Whilst I agree the "solar radiation management" could be argued to be a little outdated, the updated SRM terminology "Solar Radiation Modification" is central to the categorizations within both the recent 2022 WMO/UNEP Scientific Assessment of Ozone Depletion report and the WCRP lighthouse activity (e.g. https://www.wcrp-climate.org/ci-overview ).

Geoengineering is clearly a controversial topic, and whilst I was not involved in the planning of either activity, whereas the term "climate intervention" is central to the WCRP activity, it is noticeable that the term "climate intervention" does not feature ( from what I can see ) in the 2022 WMO/UNEP ozone assessment text (e.g. https://csl.noaa.gov/assessments/ozone/2022/executivesummary/#section-5 )

Clearly the ozone assessment focuses on the ozone layer, and this difference may well not be significant, at likely simply reflects the decision of the author-teams and leadership groups as to what might be best, considering the topic is clearly quite controversial.

And whilst I expect some might well argue the framing "climate intervention" could be considered by some to associate geoengineering in too positive or accepted a framing, others might contend that's not at all the case, the word "intervention" potentially having either positive or negative association.

The point I am trying to make here is it's clearly a controversial topic, and it's also clear that choices of terminology can understandably trigger differing sensitivities across different communities (based on experiences or feelings).

The manuscript the authors submitted features both "climate intervention" and SRM.
And given that SRM features in both the recent WCRP and WMO/UNEP reports, and the recent intercomparison, I think on this occasion, the reviewer may be mistaken that the SRM term is confusing.

It's clear there's obviously a diversity of opinions on this controversial topic, but

given also there may well be sensitivity to either term, replacing all 6 instances of SRM to CI in response to 1 reviewer's comments seems inconsistent with the manuscript the 4 scientists reviewed.

All the above said, this remains a minor typo-revision, to re-instate the original instances of "solar radiation management", the typo-edit to change "management" to the updated term "solar radiation modification".

The change then I'm requesting is:

1.1) Delete the 1st of the 2 instances of "Climate intervention" in the first sentence of the Abstract, re-instating to "Solar radiation modification" rather than "... management".

Thanks, we corrected this.

1.2) Please re-instate the instances of "(strat-SRM)" on lines 2, 3, 4, 7 and 11. This wording was present in the manuscript reviewed, and I'm not sure why one of the 4 considered confusing. It seems to me a reasonable abbreviation, to be clear this is stratospheric SRM (distinct from marine cloud brightening SRM for example). And for example the deletion on line 4 then loses the specificity of that particular model intercomparison. The previous text seemed more balanced, the term SRM aligning for example with both the WCRP and WMO/UNEP activities.

We replaced all occurrences of climate intervention (except line 1 in Abstract) by "strat-SRM". We also adapted line 35 in the manuscript accordingly. People who will be reading this manuscript will understand what we are talking about anyway.

The reviewer's assertion this is "confusing" is not correct, and reverting to the above is clearer, for example also on line 3 then clear this is stratospheric SRM rather than other climate intervention technologies.

2) Line 437 – grammatical error here – please delete "they" (before "would occur"). The new text says "as they would occur in climate intervention scenarios", but the English grammar is best stated "as would occur in climate intervention scenarios"

Thanks, we corrected this.

3) Line 445 – please change "the use of M7 with lognormal modes results in a minimum..."
Instead to "the use of M7 with lognormal modes can result in a minimum..."

The minimum only occurs when the two modes have similar magnitude concentrations, have a difference in size (to then not be overlapping). And considering also that one mode much higher number of particles than the other, the wording "can result in" is more correct.

Thanks, we corrected this.

4) Line 504 – the term "numerical stability" is not really the issue here. Numerical stability tends to refer to an iterative integration method incorporating a particular algorithmic difference-equation solution method. In this case the text is referring simply to the number of timesteps, and then a simpler issue of the approximation. It's true that changing the timestep can make an algorithm unstable or introduce numerical stability issues, but the context here is not discussing that, it's referring to the simplified process-split methods many of the microphysics modules currently use.
Please change "focusing on the numerical stability" to "including to explore differences among the process-split sub-stepping methods" or similar but slightly reduced words.
Thanks, we corrected this.

5) Line 504 – insert the word "schemes" between "of aerosol microphysics" and "under conditions".

Thanks, we corrected this.

6) Line 513 – delete "small-scale field studies" and suggest to replace instead with "co-ordinated model intercomparison" or similar statement that then aligns with the manuscript's research.

Thanks, we corrected this.

Making any statement about the question of whether small-scale field studies could be beneficial is beyond the scope of this manuscript. And although this was not queried by any of the 4 reviewers, there was quite some controversy for example of the recent SCOPEX planned activity in Sweden, and with the SPICE consortium. Any statement here is clearly beyond the scope of this article's research topic, and then should not feature in this closing sentence.

7) Line 514 – I think the "improve existing numerical models" is here referring to steering to encourage advocating for research projects to include to focus also on improving the algorithmic or sub-stepping methods within interactive stratospheric

aerosol models. Perhaps the authors would argue the types of solvers present in chemical integration methods should consider to align also with aerosol tracers, or allocating some effort/funding towards progression to dedicated aerosol "solvers" within these models? Can the reviewers be more specific here?

We have changed to: "Furthermore, additional laboratory and co-ordinated model intercomparison studies of aerosol formation, growth and dispersion under various stratospheric conditions could also be beneficial to evaluate and improve existing numerical models or to develop new explicit aerosol schemes, which potentially will be directly integrated in chemical solvers."

The specific types of improvements which could be implemented were discussed in detail in the conclusion section.

References

Weisenstein et al. (2022), "An interactive stratospheric aerosol model intercomparison of solar geoengineering by stratospheric injection of SO2 or accumulation-mode sulfuric acid aerosols",
Atmos. Chem. Phys., 22, 2955–2973, https://doi.org/10.5194/acp-22-2955-2022

---

## Author Response (AR3)

Dear Dr. Graham,

Thank you very much for your suggestions on improving the abstract as well as the final sentence. With "explicit aerosol schemes" we meant the explicit calculation of microphysics instead of "implicit calculations with operator splitting" (i.e. with sub-stepping). But we see that this could be mistaken as it was written.

We have implemented all your suggestions as proposed except for a slight adaption in the final sentence of the paper:

"This study has shown that technical developments of the models can improve the fidelity of strat-SRM assessments, and motivates dedicated effort towards further developing existing aerosol schemes for more sophisticated numerical methods, including potentially incorporating aerosol tracer tendencies into existing gas phase chemical solvers."

We think the word "assessment" is more suitable instead of the word "predictions", since the word "prediction" could be mistaken for implying future deployment of strat-SRM.

Sincerely,

Andrea Stenke, Sandro Vattioni & Co-Authros